# Regulation of store-operated Ca²⁺ entry by IP₃ receptors independent of their ability to release Ca²⁺

Pragnya Chakraborty[1,2], Bipan Kumar Deb[1†], Vikas Arige[3], Thasneem Musthafa[1], Sundeep Malik[3], David I Yule[3], Colin W Taylor[4*], Gaiti Hasan[1*‡]

[1]National Centre for Biological Sciences, Tata Institute of Fundamental Research, Bangalore, India; [2]SASTRA University, Thanjavur, India; [3]Department of Pharmacology and Physiology, University of Rochester, Rochester, United States; [4]Department of Pharmacology, University of Cambridge, Cambridge, United Kingdom

*For correspondence:
cwt1000@cam.ac.uk (CWT);
gaiti@ncbs.res.in (GH)

Present address: †Department of Molecular and Cell Biology, University of California, Berkeley, United States

‡Lead Contact

**Abstract** Loss of endoplasmic reticular (ER) Ca²⁺ activates store-operated Ca²⁺ entry (SOCE) by causing the ER localized Ca²⁺ sensor STIM to unfurl domains that activate Orai channels in the plasma membrane at membrane contact sites (MCS). Here, we demonstrate a novel mechanism by which the inositol 1,4,5 trisphosphate receptor (IP₃R), an ER-localized IP₃-gated Ca²⁺ channel, regulates neuronal SOCE. In human neurons, SOCE evoked by pharmacological depletion of ER-Ca²⁺ is attenuated by loss of IP₃Rs, and restored by expression of IP₃Rs even when they cannot release Ca²⁺, but only if the IP₃Rs can bind IP₃. Imaging studies demonstrate that IP₃Rs enhance association of STIM1 with Orai1 in neuronal cells with empty stores; this requires an IP₃-binding site, but not a pore. Convergent regulation by IP₃Rs, may tune neuronal SOCE to respond selectively to receptors that generate IP₃.

## Editor's evaluation

This paper proposes a fundamental new role for IP3 receptors in the regulation of store-operated calcium entry in neurons, in which IP3-bound receptors enhance the association of STIM1 and Orai1 independently of their ability to release Ca from the ER. While the evidence for this phenomenon is solid, experimental support for an underlying mechanism is incomplete and will require additional studies. The paper will appeal to cell biologists and neurobiologists interested in calcium signaling pathways, particularly store-operated calcium entry.

## Introduction

The activities of all eukaryotic cells are regulated by increases in cytosolic-free Ca²⁺ concentration ($[Ca^{2+}]_c$), which are almost invariably evoked by the opening of Ca²⁺-permeable ion channels in biological membranes. The presence of these Ca²⁺ channels within the plasma membrane (PM) and the membranes of intracellular Ca²⁺ stores, most notably the endoplasmic reticulum (ER), allows cells to use both intracellular and extracellular sources of Ca²⁺ to evoke Ca²⁺ signals. In animal cells, the most widely expressed Ca²⁺ signaling sequence links extracellular stimuli, through their specific receptors and activation of phospholipase C, to formation of inositol 1,4,5-trisphosphate (IP₃), which then stimulates Ca²⁺ release from the ER through IP₃ receptors (IP₃R) (*Foskett et al., 2007*; *Prole and Taylor, 2019*). IP₃Rs occupy a central role in Ca²⁺ signaling by releasing Ca²⁺ from the ER. IP₃Rs thereby elicit cytosolic Ca²⁺ signals, and by depleting the ER of Ca²⁺ they initiate a sequence that leads to activation of store-operated Ca²⁺ entry (SOCE) across the PM (*Putney, 1986*; *Thillaiappan et al., 2019*). SOCE

occurs when loss of $Ca^{2+}$ from the ER causes $Ca^{2+}$ to dissociate from the luminal $Ca^{2+}$-binding sites of an integral ER protein, stromal interaction molecule 1 (STIM1). STIM1 then unfolds its cytosolic domains to expose a region that binds directly to a $Ca^{2+}$ channel within the PM, Orai, causing it to open and $Ca^{2+}$ to flow into the cell across the PM (*Parekh and Putney, 2005*; *Prakriya and Lewis, 2015*; *Lewis, 2020*). The interactions between STIM1 and Orai occur across a narrow gap between the ER and PM, a membrane contact site (MCS), where STIM1 puncta trap Orai channels. While STIM1 and Orai are undoubtedly the core components of SOCE, many additional proteins modulate their interactions (*Rosado et al., 2000*; *Palty et al., 2012*; *Deb et al., 2016*; *Srivats et al., 2016*) and other proteins contribute by regulating the assembly of MCS (*Chang et al., 2013*; *Giordano et al., 2013*; *Kang et al., 2019*).

It is accepted that $IP_3$-evoked $Ca^{2+}$ release from the ER through $IP_3$Rs is the usual means by which extracellular stimuli evoke SOCE. Here, the role of the $IP_3$R is widely assumed to be restricted to its ability to mediate $Ca^{2+}$ release from the ER and thereby activate STIM1. Evidence from *Drosophila*, where we suggested an additional role for $IP_3$Rs in regulating SOCE (*Agrawal et al., 2010*; *Chakraborty et al., 2016*), motivated the present study, wherein we examined the contribution of $IP_3$Rs to SOCE in mammalian neurons. We show that in addition to their ability to activate STIM1 by evoking ER $Ca^{2+}$ release, $IP_3$Rs also facilitate interactions between active STIM1 and Orai1. This additional role for $IP_3$Rs, which is regulated by $IP_3$ but does not require a functional pore, reveals an unexpected link between $IP_3$, $IP_3$Rs and $Ca^{2+}$ signaling that is not mediated by $IP_3$-evoked $Ca^{2+}$ release. We speculate that dual regulation of SOCE by $IP_3$Rs may allow $Ca^{2+}$ release evoked by $IP_3$ to be preferentially coupled to SOCE.

## Results

### Loss of $IP_3$R1 attenuates SOCE in human neural stem cells and neurons

We investigated the effects of $IP_3$Rs on SOCE by measuring $[Ca^{2+}]_c$ in human neural stem cells and neurons prepared from embryonic stem cells. Human neural progenitor cells (hNPCs) were derived from H9 embryonic stem cells using small molecules that mimic cues provided during human brain development (*Gopurappilly et al., 2018*). We confirmed that hNPCs express canonical markers of neural stem cells (*Figure 1A*) and that $IP_3$R1 is the predominant $IP_3$R subtype (GEO accession no. GSE109111; *Gopurappilly et al., 2018*). An inducible lentiviral shRNA-miR construct targeting $IP_3$R1 reduced $IP_3$R1 expression by 93 ± 0.4% relative to a non-silencing (NS) construct (*Figure 1B and C*). Carbachol stimulates muscarinic acetylcholine receptors, which are expressed at low levels in hNPCs (*Gopurappilly et al., 2018*). In $Ca^{2+}$-free medium, carbachol evoked an increase in $[Ca^{2+}]_c$ in about 10% of hNPCs, consistent with it stimulating $Ca^{2+}$ release from the ER through $IP_3$Rs. Restoration of extracellular $Ca^{2+}$ then evoked an increase in $[Ca^{2+}]_c$ in all cells that responded to carbachol. Both carbachol-evoked $Ca^{2+}$ release and SOCE were abolished in hNPCs expressing $IP_3$R1-shRNA, confirming the effectiveness of the $IP_3$R1 knockdown (*Figure 1—figure supplement 1A–C*).

Thapsigargin, a selective and irreversible inhibitor of the ER $Ca^{2+}$ pump (sarcoplasmic/endoplasmic reticulum $Ca^{2+}$-ATPase, SERCA), was used to deplete the ER of $Ca^{2+}$ and thereby activate SOCE (*Figure 1D*; *Parekh and Putney, 2005*). Restoration of extracellular $Ca^{2+}$ to thapsigargin-treated hNPCs evoked a large increase in $[Ca^{2+}]_c$, reflecting the activity of SOCE (*Figure 1D*). The maximal amplitude and rate of SOCE were significantly reduced in cells lacking $IP_3$R1, but the resting $[Ca^{2+}]_c$ and thapsigargin-evoked $Ca^{2+}$ release were unaffected (*Figure 1D–F* and *Figure 1—figure supplement 1D and E*). STIM1 and Orai1 expression were also unaltered in hNPC lacking $IP_3$R1 (*Figure 1—figure supplement 1G*). After spontaneous differentiation of hNPC, cells expressed markers typical of mature neurons, and the cells responded to depolarization with an increase in $[Ca^{2+}]_c$ (*Figure 1—figure supplement 1F* and *Figure 1—figure supplement 1H–J*). Thapsigargin evoked SOCE in these differentiated neurons; and expression of $IP_3$R1-shRNA significantly reduced the SOCE response without affecting depolarization-evoked $Ca^{2+}$ signals (*Figure 1H–J* and *Figure 1—figure supplement 1H–L*).

### Loss of $IP_3$R1 attenuates SOCE in human neuroblastoma cells

$IP_3$Rs link physiological stimuli that evoke $Ca^{2+}$ release from the ER to SOCE, but the contribution of $IP_3$Rs is thought to be limited to their ability to deplete the ER of $Ca^{2+}$. We have reported that in *Drosophila* neurons there is an additional requirement for $IP_3$Rs independent of ER $Ca^{2+}$ release

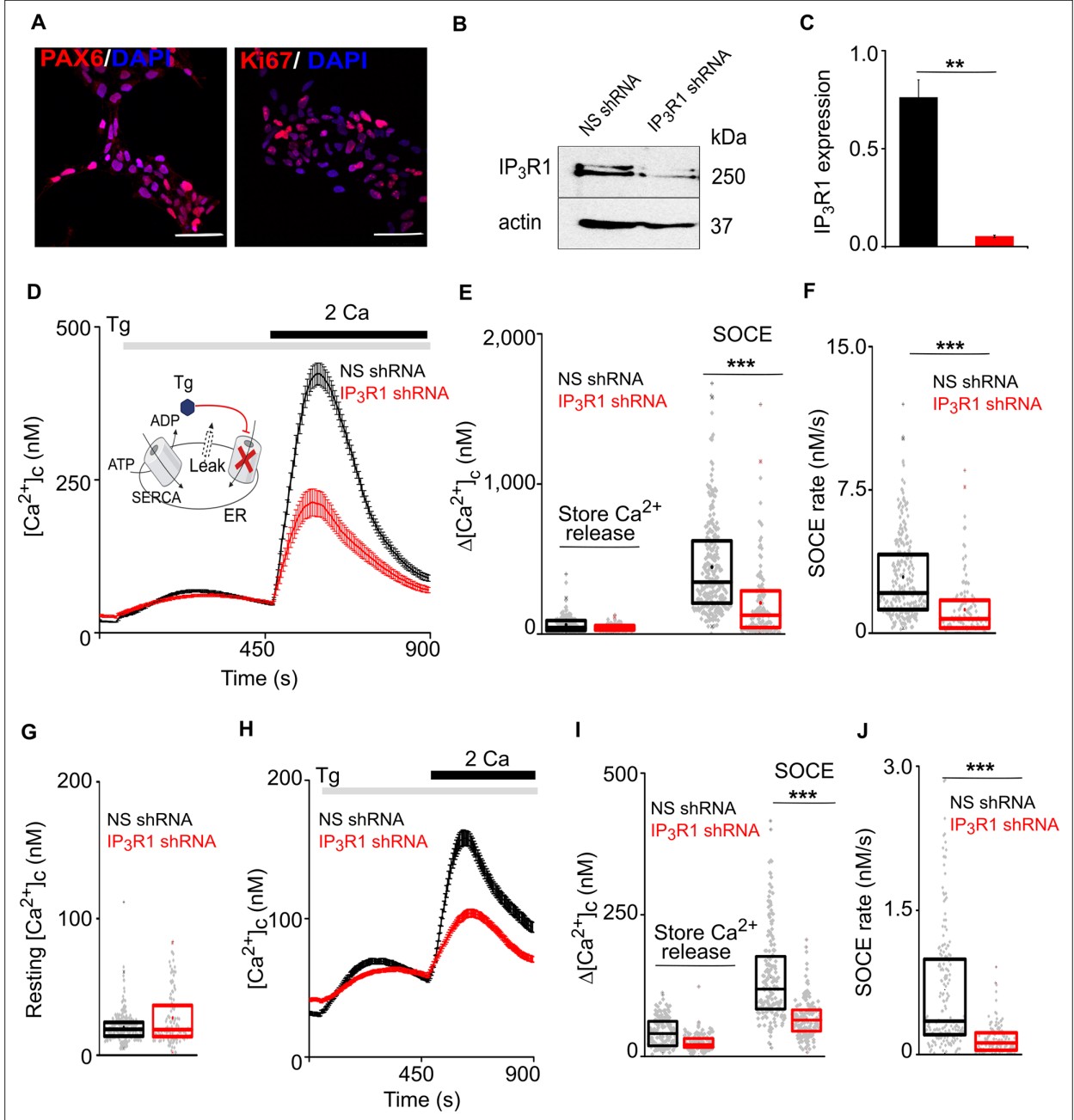

**Figure 1.** Loss of IP₃R1 attenuates SOCE in human neural stem cells. (**A**) Confocal images of hNPCs (passage 6) stained for DAPI and neural stem cell proteins: Pax6 and Ki67 (proliferation marker). Scale bars, 50 μm. (**B**) WB for IP₃R1 of hNPCs expressing non-silencing (NS) or IP₃R1-shRNA. (**C**) Summary results (mean ±s.d., n=3) show IP₃R1 expression relative to actin. $^{**}p < 0.01$, Student's *t*-test with unequal variances. (**D**) Changes in [Ca²⁺]c evoked by thapsigargin (Tg, 10 μM) in Ca²⁺-free HBSS and then restoration of extracellular Ca²⁺ (2 mM) in hNPCs expressing NS or IP₃R1-shRNA. Mean ± s.e.m. from hree independent experiments, each with four replicates that together included 100–254 cells. Inset shows the target of Tg. (**E–G**) Summary results (individual cells, median (bar), 25th and 75th percentiles (box) and mean (circle)) show Ca²⁺ signals evoked by Tg or Ca²⁺ restoration (**E**), rate of Ca²⁺ entry (**F**) and resting [Ca²⁺]c (**G**). $^{***}p < 0.001$, Mann-Whitney U-test. (**H**) Changes in [Ca²⁺]c evoked by Tg (10 μM) in Ca²⁺-free HBSS and after restoring extracellular Ca²⁺ (2 mM) in neurons (differentiated hNPCs) expressing NS or IP₃R1-shRNA. Mean ± s.e.m. from three experiments with ~200 cells. (**I,J**) Summary results (presented as in E-G) show Ca²⁺ signals evoked by Tg or Ca²⁺ restoration (**I**) and rate of Ca²⁺ entry (**J**). $^{***}p < 0.001$. Mann-Whitney U-test. See also *Figure 1—figure supplement 1*. Source data in *Figure 1—source data 1*.

The online version of this article includes the following source data and figure supplement(s) for figure 1:

**Source data 1.** Loss of IP₃R1 attenuates SOCE in human neural stem cells.

**Figure supplement 1.** Loss of IP₃R1 attenuates SOCE in neural precursor cells and differentiated neurons.

**Figure supplement 1—source data 1.** Loss of IP₃R1 attenuates SOCE in neural precursor cells and differentiated neurons.

(*Venkiteswaran and Hasan, 2009*; *Agrawal et al., 2010*; *Chakraborty et al., 2016*). Our results with hNPCs and stem cell-derived neurons suggest a similar requirement for IP$_3$Rs in regulating SOCE in mammalian neurons. To explore the mechanisms underlying this additional role for IP$_3$Rs, we turned to a more tractable cell line, SH-SY5Y cells. These cells are derived from a human neuroblastoma; they exhibit many neuronal characteristics (*Agholme et al., 2010*); they express M3 muscarinic acetylcholine receptors that evoke IP$_3$-mediated Ca$^{2+}$ release and SOCE (*Grudt et al., 1996*); and they express predominantly IP$_3$R1 (*Wojcikiewicz, 1995*; *Tovey et al., 2001*), with detectable IP$_3$R3, but no IP$_3$R2 (*Figure 2A*). We used inducible expression of IP$_3$R1-shRNA to significantly reduce IP$_3$R1 expression (by 74 ± 1.2%), without affecting IP$_3$R3 (*Figure 2A and B*). As expected, carbachol-evoked Ca$^{2+}$ signals in individual SH-SY5Y cells were heterogenous and the carbachol-evoked Ca$^{2+}$ release was significantly reduced by knockdown of IP$_3$R1 (*Figure 2C and D* and *Figure 2—figure supplement 1A and B*). Thapsigargin evoked SOCE in SH-SY5Y cells (*Grudt et al., 1996*), and it was significantly attenuated after knockdown of IP$_3$R1 without affecting resting [Ca$^{2+}$]$_c$, the Ca$^{2+}$ release evoked by thapsigargin or expression of STIM1 and Orai1 (*Figure 2E–G* and *Figure 2—figure supplement 1C–E*).

We also used CRISPR/Cas9n and Cas9 to disrupt one or both copies of the IP$_3$R1 gene, subsequently referred to as IKO (one copy knockout) and IKO null (both copies knocked out) in SH-SY5Y cells. IP$_3$R1 expression was absent in the IKO null (*Figure 2—figure supplement 1F*) whereas expression of STIM1, STIM2 and Orai1 were unperturbed (*Figure 2—figure supplement 1G*). Carbachol-evoked Ca$^{2+}$ release and thapisgargin-evoked SOCE were significantly reduced (*Figure 2—figure supplement 1H–J*). Since the IKO null cells were fragile and grew slowly, we examined SOCE in SH-SY5Y cells with disruption of one copy of the IP$_3$R1 gene. In the IKO cells, IP$_3$R1 expression, carbachol-evoked Ca$^{2+}$ signals and thapsigargin-evoked SOCE were all reduced (*Figure 2—figure supplement 1K–Q*).

These observations, which replicate those from hNPCs and neurons (*Figure 1*), vindicate our use of SH-SY5Y cells to explore the mechanisms linking IP$_3$Rs to SOCE in human neurons.

Expression of IP$_3$R1 or IP$_3$R3 in SH-SY5Y cells expressing IP$_3$R1-shRNA restored both carbachol-evoked Ca$^{2+}$ release and thapsigargin-evoked SOCE without affecting resting [Ca$^{2+}$]$_c$ or thapsigargin-evoked Ca$^{2+}$ release (*Figure 2H–J* and *Figure 2—figure supplement 2A–D*). Over-expression of STIM1 in cells expressing NS-shRNA had no effect on SOCE (*Figure 2—figure supplement 2E and F*), but it restored thapsigargin-evoked SOCE in cells expressing IP$_3$R1-shRNA, without affecting resting [Ca$^{2+}$]$_c$ or thapsigargin-evoked Ca$^{2+}$ release (*Figure 2K–M*). We conclude that IP$_3$Rs are required for optimal SOCE, but they are not essential because additional STIM1 can replace the need for IP$_3$Rs (*Figure 3A*).

It has been reported that SOCE is unaffected by loss of IP$_3$R in non-neuronal cells (*Ma et al., 2001*; *Chakraborty et al., 2016*). Consistent with these observations, the SOCE evoked in HEK cells by stores emptied fully by treatment with thapsigargin was unaffected by expression of IP$_3$R1 shRNA (*Figure 2—figure supplement 3A–3C*) or by knockout of all three IP$_3$R subtypes using CRISPR/cas9 (HEK-TKO cells; *Figure 2—figure supplement 3D and E*). The association of STIM1 with Orai1 in wild type HEK cells and HEK TKO cells after thapsigargin-evoked store depletion also appeared identical as tested by a proximity ligation assay (PLA, described further in Figure 5 and *Figure 2—figure supplement 3F*). Neuronal and non-neuronal cells may, therefore, differ in the contribution of IP$_3$R to SOCE. We return to this point later.

## Binding of IP$_3$ to IP$_3$R without a functional pore stimulates SOCE

IP$_3$Rs are large tetrameric channels that open when they bind IP$_3$ and Ca$^{2+}$, but they also associate with many other proteins (*Prole and Taylor, 2019*), and many IP$_3$Rs within cells appear not to release Ca$^{2+}$ (*Thillaiappan et al., 2019*). A point mutation (D2550A, IP$_3$R1$^{D/A}$) within the IP$_3$R1 pore prevents it from conducting Ca$^{2+}$ (*Dellis et al., 2008*). As expected, expression of IP$_3$R1$^{D/A}$ in cells lacking IP$_3$R1 failed to rescue carbachol-evoked Ca$^{2+}$ release, but it unexpectedly restored thapsigargin-evoked SOCE (*Figure 3B-D*; and *Figure 3—figure supplement 1*). We confirmed that rescue of thapsigargin-evoked Ca$^{2+}$ entry by this pore-dead IP$_3$R was mediated by a conventional SOCE pathway by demonstrating that it was substantially attenuated by siRNA-mediated knockdown of Orai1 (*Figure 3C and D* and *Figure 3—figure supplement 1F–H*).

Activation of IP$_3$Rs is initiated by IP$_3$ binding to the N-terminal IP$_3$-binding core of each IP$_3$R subunit (*Prole and Taylor, 2019*). Mutation of two conserved phosphate-coordinating residues in the α-domain of the binding core (R568Q and K569Q of IP$_3$R1, IP$_3$R1$^{RQ/KQ}$) almost abolishes IP$_3$ binding

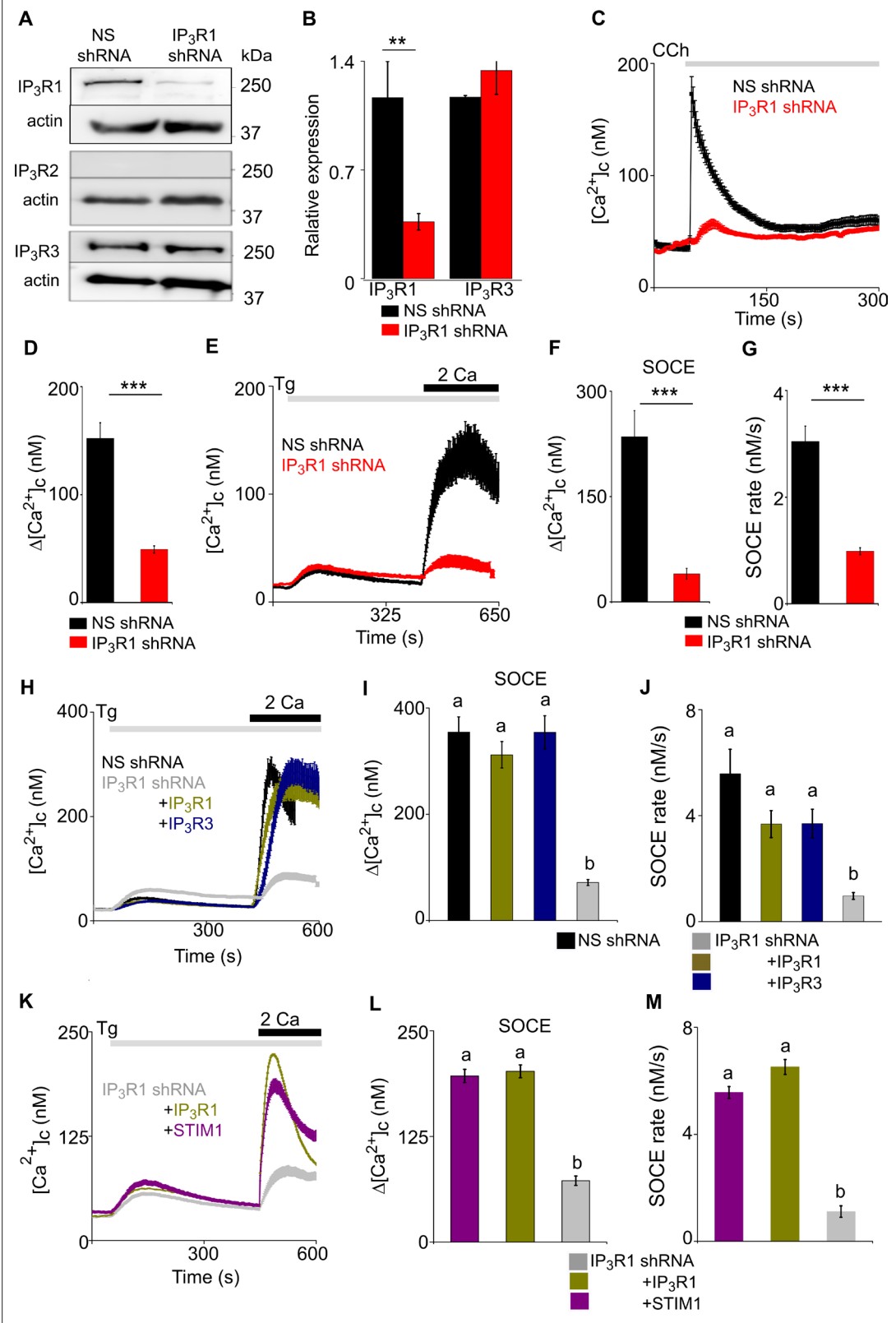

**Figure 2.** Loss of IP$_3$R1 attenuates SOCE in SH-SY5Y cells. (**A**) WB for IP$_3$R1-3 of SH-SY5Y cells expressing non-silencing (NS) or IP$_3$R1-shRNA. (**B**) Summary results (mean ± s.d., n=4) show IP$_3$R expression relative to actin normalized to control NS cells. **p < 0.01, Student's *t*-test with unequal variances. (**C**) Ca$^{2+}$ signals evoked by carbachol (CCh, 3 µM) in SH-SY5Y cells expressing NS or IP$_3$R1-shRNA. Mean ± s.e.m. from three experiments with 70–90 cells. (**D**) Summary results show peak changes in [Ca$^{2+}$]$_c$ (Δ[Ca$^{2+}$]$_c$) evoked by CCh. ***p < 0.001, Mann-Whitney U-test. (**E**) Ca$^{2+}$ signals evoked by

*Figure 2 continued*

thapsigargin (Tg, 10 µM) in $Ca^{2+}$-free HBSS and then after restoration of extracellular $Ca^{2+}$ (2 mM) in cells expressing NS or IP₃R1-shRNA. Mean ± s.e.m. from three experiments with ~50 cells. (**F, G**) Summary results (individual cells, mean ± s.e.m., n=3, ~50 cells) show peak changes in $[Ca^{2+}]_c$ evoked by $Ca^{2+}$ restoration ($\Delta[Ca^{2+}]_c$) (**F**) and rate of $Ca^{2+}$ entry (**G**). ***p < 0.001, Mann-Whitney U-test. (**H**) $Ca^{2+}$ signals evoked by Tg and then $Ca^{2+}$ restoration in cells expressing NS-shRNA, or IP₃R1-shRNA alone or with IP₃R1 or IP₃R3. Traces show mean ± s.e.m. (50–115 cells from three experiments). (**I, J**) Summary results (mean ± s.e.m, 50–115 cells from three experiments) show peak increases in $[Ca^{2+}]_c$ ($\Delta[Ca^{2+}]_c$) evoked by $Ca^{2+}$ restoration (**I**) and rates of $Ca^{2+}$ entry (**J**) evoked by restoring extracellular $Ca^{2+}$. (**K**) Effects of thapsigargin (Tg, 10 µM) in $Ca^{2+}$-free HBSS and then after $Ca^{2+}$ restoration (2 mM) in cells expressing IP₃R1-shRNA alone or with IP₃R1 or mCh-STIM1. Traces show mean ± s.e.m. (100–150 cells from three experiments). (**L, M**) Summary results (mean ± s.e.m.) show peak increase in $[Ca^{2+}]_c$ after $Ca^{2+}$ restoration ($\Delta[Ca^{2+}]_c$) (**L**) and rate of $Ca^{2+}$ entry (**M**). Different letters indicate significant differences (panels **I, J, L, M**), p <0.001, one-way ANOVA with pair-wise Tukey's test. See also *Figure 2—figure supplements 1–3*. Source data in *Figure 2—source data 1*.

The online version of this article includes the following source data and figure supplement(s) for figure 2:

**Source data 1.** Loss of IP₃R1 attenuates SOCE in SH-SY5Y cells.

**Figure supplement 1.** Reduced expression of IP₃R1 using either shRNA or CRISPR/Cas9n attenuates SOCE in SH-SY5Y cells.

**Figure supplement 1—source data 1.** Reduced expression of IP₃R1 using either shRNA or CRISPR/Cas9n attenuates SOCE in SH-SY5Y cells.

**Figure supplement 2.** Attenuated SOCE in SH-SY5Y cells lacking IP₃R1 is rescued by expression of IP₃R1, IP₃R3 or STIM1.

**Figure supplement 2—source data 1.** Attenuated SOCE in SH-SY5Y cells lacking IP₃R1 is rescued by expression of IP₃R1, IP₃R3 or STIM1.

**Figure supplement 3.** Loss of IP₃R1 does not affect SOCE in HEK cells.

**Figure supplement 3—source data 1.** Loss of IP₃R1 does not affect SOCE in HEK cells.

(*Yoshikawa et al., 1996*; *Iwai et al., 2007*), while mutation of a single residue (R568Q, IP₃R1$^{RQ}$) reduces the IP₃ affinity by ~10-fold (*Dellis et al., 2008*). Expression of rat IP₃R1$^{RQ/KQ}$ rescued neither carbachol-evoked $Ca^{2+}$ release nor thapsigargin-evoked SOCE in cells lacking IP₃R1 (*Figure 3E and F* and *Figure 3—figure supplement 1C and I*). However, expression of IP₃R1$^{RQ}$ substantially rescued thapsigargin-evoked SOCE (*Figure 3E and F* and *Figure 3—figure supplement 1J*). Expression of an N-terminal fragment of rat IP₃R (IP₃R1$^{1-604}$), to which IP₃ binds normally (*Iwai et al., 2007*), failed to rescue thapsigargin-evoked SOCE (*Figure 3—figure supplement 1K and L*). These results establish that a functional IP₃-binding site within a full-length IP₃R is required for IP₃Rs to facilitate thapsigargin-evoked SOCE. Hence in cells with empty $Ca^{2+}$ stores, IP₃ binding, but not pore-opening, is required for regulation of SOCE by IP₃Rs. In cells stimulated only with thapsigargin and expressing IP₃Rs with deficient IP₃ binding, basal levels of IP₃ are probably insufficient to meet this need.

We further examined the need for IP₃ by partially depleting the ER of $Ca^{2+}$ using cyclopiazonic acid (CPA), a reversible inhibitor of SERCA, to allow submaximal activation of SOCE (*Figure 3—figure supplement 1M and N*). Under these conditions, addition of carbachol in $Ca^{2+}$-free HBSS to SH-SY5Y cells expressing IP₃R1-shRNA caused a small increase in $[Ca^{2+}]_c$ (*Figure 4A–C*). In the same cells expressing IP₃R1$^{DA}$, the carbachol-evoked $Ca^{2+}$ release was indistinguishable from that observed in cells without IP₃R$^{DA}$ (*Figure 4B and C*), indicating that the small response was entirely mediated by residual native IP₃R1 and/or IP₃R3. Hence, the experiment allows carbachol to stimulate IP₃ production in cells expressing IP₃R1$^{DA}$ without causing additional $Ca^{2+}$ release. The key result is that in cells expressing IP₃R1$^{DA}$, carbachol substantially increased SOCE from sub maximal to higher levels (*Figure 4AC*). Moreover, addition of carbachol to control shRNA expressing SH-SY5Y cells with maximal store depletion (thapsigargin, Tg, 2 µM) resulted in a small increase in SOCE (*Figure 4—figure supplement 1A*). We conclude that in neuronal cells IP₃, through IP₃Rs, regulates coupling of empty stores to SOCE. This is the first example of an IP₃R mediating a response to IP₃ that does not require the pore of the channel.

G-protein-coupled receptors are linked to IP₃ formation through the G-protein Gq, which stimulates phospholipase C β (PLC β). We used YM-254890 to inhibit Gq (*Kostenis et al., 2020*; *Patt et al., 2021*). As expected, addition of YM-254890 to wild type (WT) or NS-shRNA transfected SH-SY5Y cells abolished the $Ca^{2+}$ signals evoked by carbachol (*Figure 4—figure supplement 1C*), but it also reduced the maximal amplitude and rate of thapsigargin-evoked SOCE (*Figure 4D–E* and *Figure 3—figure supplement 1O*). YM-254890 had no effect on the residual thapsigargin-evoked SOCE in SH-SY5Y cells expressing IP₃R1-shRNA (*Figure 4F* and *Figure 3—figure supplement 1O*). The latter result is important because it demonstrates that the inhibition of SOCE in cells with functional IP₃Rs is not an off-target effect causing a direct inhibition of SOCE.

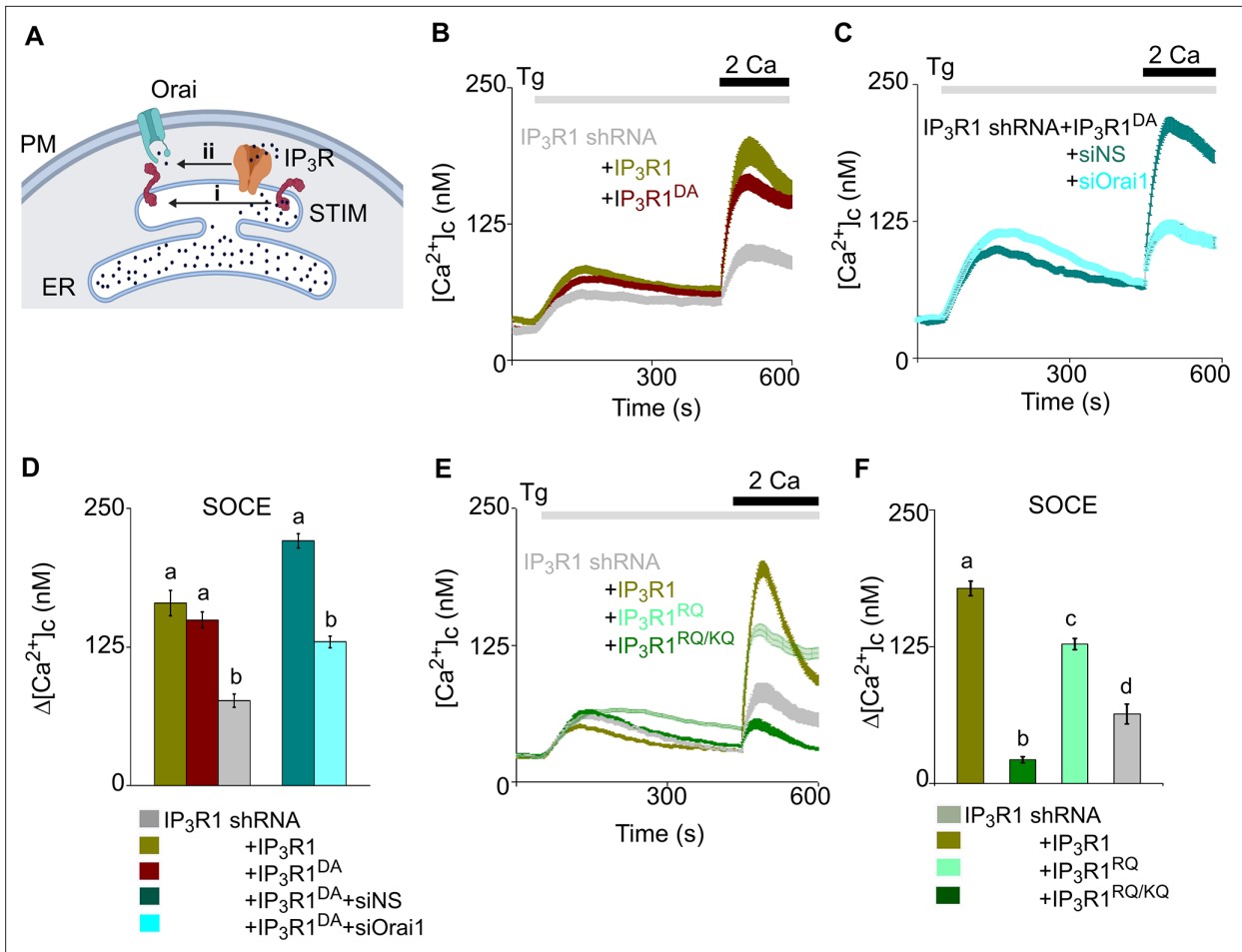

**Figure 3.** Regulation of SOCE by IP₃R requires IP₃ binding but not a functional pPore in SH-SY5Y cells. (**A**) SOCE is activated when loss of Ca²⁺ from the ER through IP₃Rs activates STIM1 (**i**). Our results suggest an additional role for IP₃Rs (**ii**). (**B**) SH-SY5Y cells expressing IP₃R1-shRNA alone or with IP₃R1 or IP₃R1$^{DA}$ were stimulated with thapsigargin (Tg, 1 µM) in Ca²⁺-free HBSS before restoring extracellular Ca²⁺ (2 mM). Traces show mean ± s.e.m, for 100–150 cells from three experiments. (**C**) Cells expressing IP₃R1-shRNA and IP₃R1$^{DA}$ were treated with NS-siRNA or Orai1-siRNA before measuring Tg-evoked Ca²⁺ entry. Traces show mean ± s.e.m. for 85–100 cells from three experiments. (**D**) Summary results (mean ± s.e.m.) show peak increases in [Ca²⁺]$_c$ (Δ[Ca²⁺]$_c$) evoked by Ca²⁺ restoration. (**E**) Tg-evoked Ca²⁺ entry in cells expressing IP₃R1-shRNA with IP₃R1, IP₃R1$^{RQ}$ or IP₃R1$^{RQ/KQ}$. Traces show mean ± s.e.m, for 90–150 cells from three experiments. (**F**) Summary results (mean ± s.e.m.) show peak increases in [Ca²⁺]$_c$ (Δ[Ca²⁺]$_c$) evoked by Ca²⁺ restoration. Different letter codes (panels **D**, **F**) indicate significantly different values, p<0.001, for multiple comparison one-way ANOVA and pair-wise Tukey's test and for two genotype comparison Mann Whitney U-test. See also *Figure 3—figure supplement 1—source data 1*. Source data in *Figure 3—source data 1*.

The online version of this article includes the following source data and figure supplement(s) for figure 3:

**Source data 1.** Regulation of SOCE by IP₃R requires IP₃ binding but not a functional pPore in SH-SY5Y cells.

**Figure supplement 1.** Attenuated SOCE in SH-SY5Y cells lacking IP₃R1 is rescued by expression of pore-dead IP₃R1 with a functional IP₃-binding site.

**Figure supplement 1—source data 1.** Attenuated SOCE in SH-SY5Y cells lacking IP₃R1 is rescued by expression of pore-dead IP₃R1 with a functional IP₃-binding site.

In wild type or HEK-TKO (lacking all three IP₃Rs) cells, YM-254890 had no effect on thapsigargin-evoked SOCE, but it did inhibit SOCE in HEK cells lacking only IP₃R1 (*Figure 4G–I* and *Figure 4—figure supplement 1D–G*). These results suggest that in HEK cells, which normally express all three IP₃R subtypes (*Mataragka and Taylor, 2018*), neither loss of IP₃R1 nor inhibition of Gαq is sufficient on its own to inhibit thapsigargin-evoked SOCE, but when combined there is a synergistic loss of SOCE.

## IP₃Rs promote interaction of STIM1 with Orai1 within MCS

Our evidence that IP₃Rs intercept coupling between empty stores and SOCE (*Figure 3A*) prompted us to investigate the coupling of STIM1 with Orai1 across the narrow junctions between ER and PM

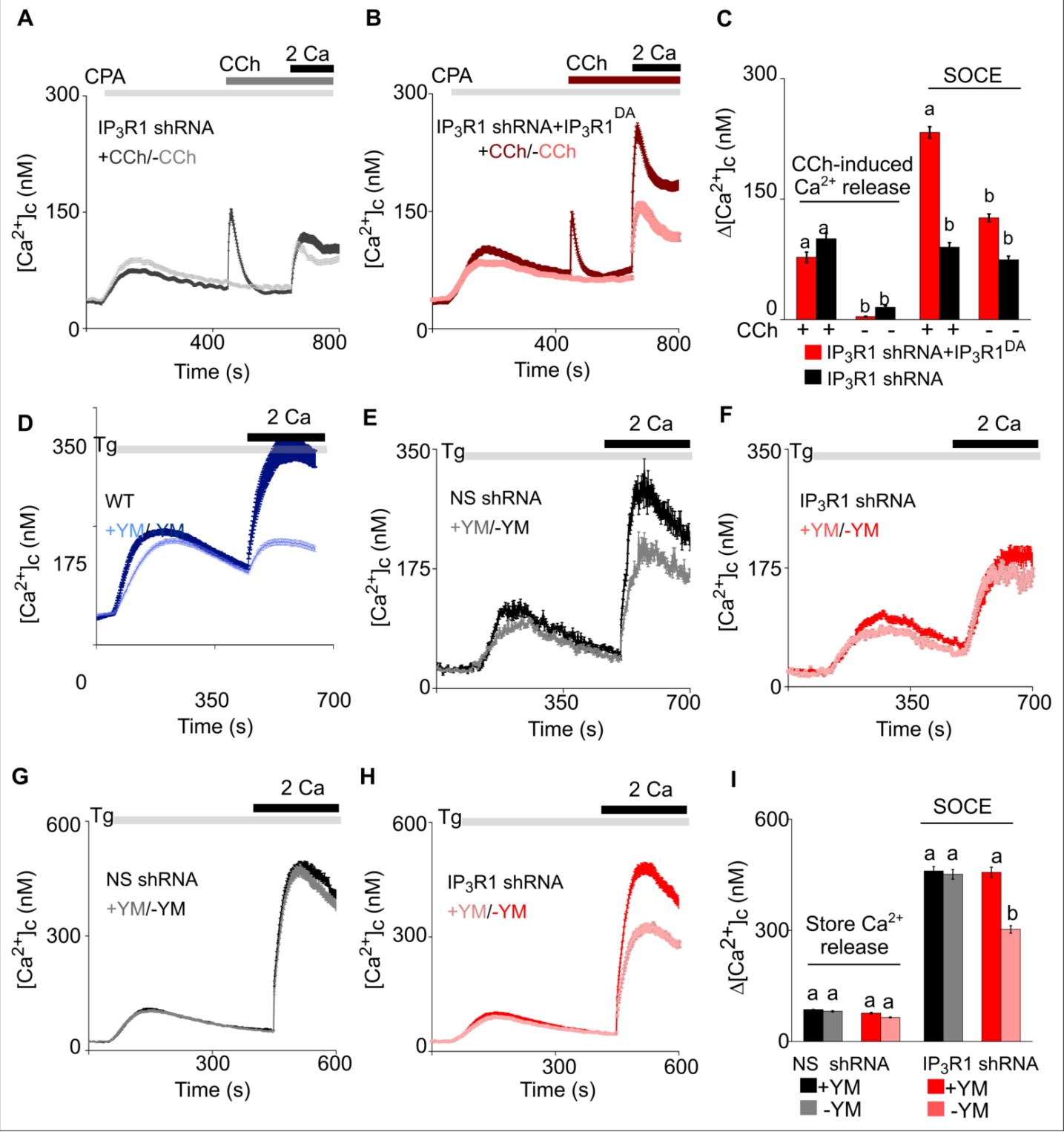

**Figure 4.** Receptor-regulated IP₃ production stimulates SOCE in cells with empty Ca²⁺ stores and expressing pore-dead IP₃R. (**A, B**) SH-SY5Y cells expressing IP₃R1-shRNA alone (**A**) or with IP₃R1$^{DA}$ (**B**) were treated with a low concentration of CPA (2 μM) in Ca²⁺-free HBSS to partially deplete the ER of Ca²⁺ and sub-maximally activate SOCE (see *Figure 3—figure supplement 1M–N*). Carbachol (CCh, 1 μM) was then added to stimulate IP₃ formation through muscarinic receptors, and extracellular Ca²⁺ (2 mM) was then restored. Traces (mean ± s.e.m of 68–130 cells from three experiments) show responses with and without the CCh addition. (**C**) Summary results show the peak increases in [Ca²⁺]$_c$ (Δ[Ca²⁺]$_c$) after addition of CCh (CCh-induced Ca²⁺ release) and then after restoring extracellular Ca²⁺ (SOCE). (**D–F**) SH-SY5Y cells wild type (WT) (**D**) and expressing NS-shRNA (**E**) or IP₃R1-shRNA (**F**) were treated with YM-254890 (YM, 1 μM, 5 min) in Ca²⁺-free HBSS to inhibit Gαq and then with thapsigargin (Tg, 1 μM) before restoring extracellular Ca²⁺

*Figure 4 continued on next page*

*Figure 4 continued*

(2 mM). Traces show mean ± s.e.m of ~120 cells from three experiments. (**G–I**) Similar analyses of HEK cells. Summary results (mean ± s.e.m, 50–100 cells from three experiments) are shown in (**I**). Different letter codes (panels C and I) indicate significantly different values within the store $Ca^{2+}$ release or SOCE groups, p<0.001, one-way ANOVA and pair-wise Tukey's test. See also *Figure 4—figure supplement 1*. Source data in *Figure 4—source data 1*.

The online version of this article includes the following source data and figure supplement(s) for figure 4:

**Source data 1.** Receptor-regulated $IP_3$ production stimulates SOCE in cells with empty $Ca^{2+}$ stores and expressing pore-dead $IP_3R$.

**Figure supplement 1.** Effects of generating $IP_3$ and inhibiting Gq on $Ca^{2+}$ signals and STIM1-Orai1 interactions in SH-SY5Y and HEK cells.

**Figure supplement 1—source data 1.** Effects of generating $IP_3$ and inhibiting Gq on $Ca^{2+}$ signals and STIM1-Orai1 interactions in SH-SY5Y and HEK cells.

(*Carrasco and Meyer, 2011*). An in situ proximity ligation assay (PLA) is well suited to analyzing this interaction because it provides a signal when two immunolabeled proteins are within ~40 nm of each other (*Derangère et al., 2016*), a distance comparable to the dimensions of the junctions wherein STIM1 and Orai1 interact (*Poteser et al., 2016*). We confirmed the specificity of the PLA and demonstrated that it reports increased association of STIM1 with Orai1 after treating SH-SY5Y cells with thapsigargin by measuring the surface area of PLA spots (*Figure 5A* and *Figure 5—figure supplement 1A–F*) and not the number, because the latter did not change upon store-depletion (*Figure 5—figure supplement 1O*). In cells expressing $IP_3R1$-shRNA, thapsigargin had no effect on the STIM1-Orai1 interaction reported by PLA, but the interaction was rescued by expression of $IP_3R1$ or $IP_3R1^{DA}$. There was no rescue with $IP_3R1^{RQ/KQ}$ (*Figure 5B–E*). WT SH-SY5Y cells that were depleted of basal $IP_3$ by treatment with the Gq inhibitor YM-254890, showed significantly reduced STIM1-Orai1 interaction after thapsigargin-evoked depletion of $Ca^{2+}$ stores (*Figure 4—figure supplement 1B*). The results with PLA exactly mirror those from functional analyses (*Figures 1–4*), suggesting that $IP_3$ binding to $IP_3R$ enhances SOCE by facilitating interaction of STIM1 with Orai1 (*Figure 3A*).

In independent experiments we tested the effect of fluorescent-tagged and ectopically expressed ligand bound (wild type rat $IP_3R1$) and mutant (rat $IP_3R1^{RQ/KQ}$; *Figure 6—figure supplement 1A*) $IP_3R1$ on SOCE dependent STIM1 oligomerization and translocation to ER-PM junctions in SH-SY5Y cells (*Figure 6*). In agreement with PLA data (*Figure 5*), ER-PM translocation of mVenus-STIM1 upon SOCE induction was reduced significantly in mCherry-$IP_3R1^{RQ/KQ}$ expressing cells compared to mCherry-$IP_3R1$ expressing SH-SY5Y cells (*Figure 6A, B, D and E* and *Figure 6—figure supplement 1B and D*). SOCE also brought about a small increase in the surface intensity of over-expressed wild type mCherry-$IP_3R1$ and mCherry-$IP_3R1^{RQ/KQ}$ in the regions where we observe formation of SOCE-dependent STIM1 puncta (*Figure 6A–C* and *Figure 6—figure supplement 1C and E*). Moreover, the intensity of mCherry-$IP_3R1^{RQ/KQ}$ appeared marginally lower than mCherry-$IP_3R1$ (*Figure 6—figure supplement 1C and E*). The significance, if any, of these small changes in surface localization between over-expressed mCherry-$IP_3R1$ and mCherry-$IP_3R1^{RQ/KQ}$ upon SOCE induction, need further verification by alternate methods.

Extended synaptotagmins (E-Syts) are ER proteins that stabilize ER-PM junctions including STIM1-Orai1 MCS (*Maléth et al., 2014*; *Kang et al., 2019*; *Woo et al., 2020*). Over-expression of E-Syt1 in SH-SY5Y cells expressing $IP_3R1$-shRNA rescued thapsigargin-evoked $Ca^{2+}$ entry without affecting resting $[Ca^{2+}]_c$ or thapsigargin-evoked $Ca^{2+}$ release (*Figure 7A–C*). The rescued $Ca^{2+}$ entry is likely to be mediated by conventional SOCE because it was substantially attenuated by knockdown of STIM1 (*Figure 7D–F*). Over-expression of E-Syt1 had no effect on SOCE in cells with unperturbed $IP_3Rs$ (*Figure 7G–I*). These results suggest that attenuated SOCE after loss of $IP_3Rs$ can be restored by exaggerating ER-PM MCS.

## Discussion

After identification of STIM1 and Orai1 as core components of SOCE (*Prakriya and Lewis, 2015*; *Thillaiappan et al., 2019*), the sole role of $IP_3Rs$ within the SOCE pathway was assumed to be the release of ER $Ca^{2+}$ that triggers STIM1 activation. The assumption is consistent with evidence that thapsigargin-evoked SOCE can occur in avian (*Sugawara et al., 1997*; *Ma et al., 2002*; *Chakraborty et al., 2016*) and mammalian cells without $IP_3Rs$ (*Prakriya and Lewis, 2001*). Although SOCE in mammalian HEK cells was unaffected by loss of $IP_3Rs$ in our study (*Figure 2—figure supplement 3*),

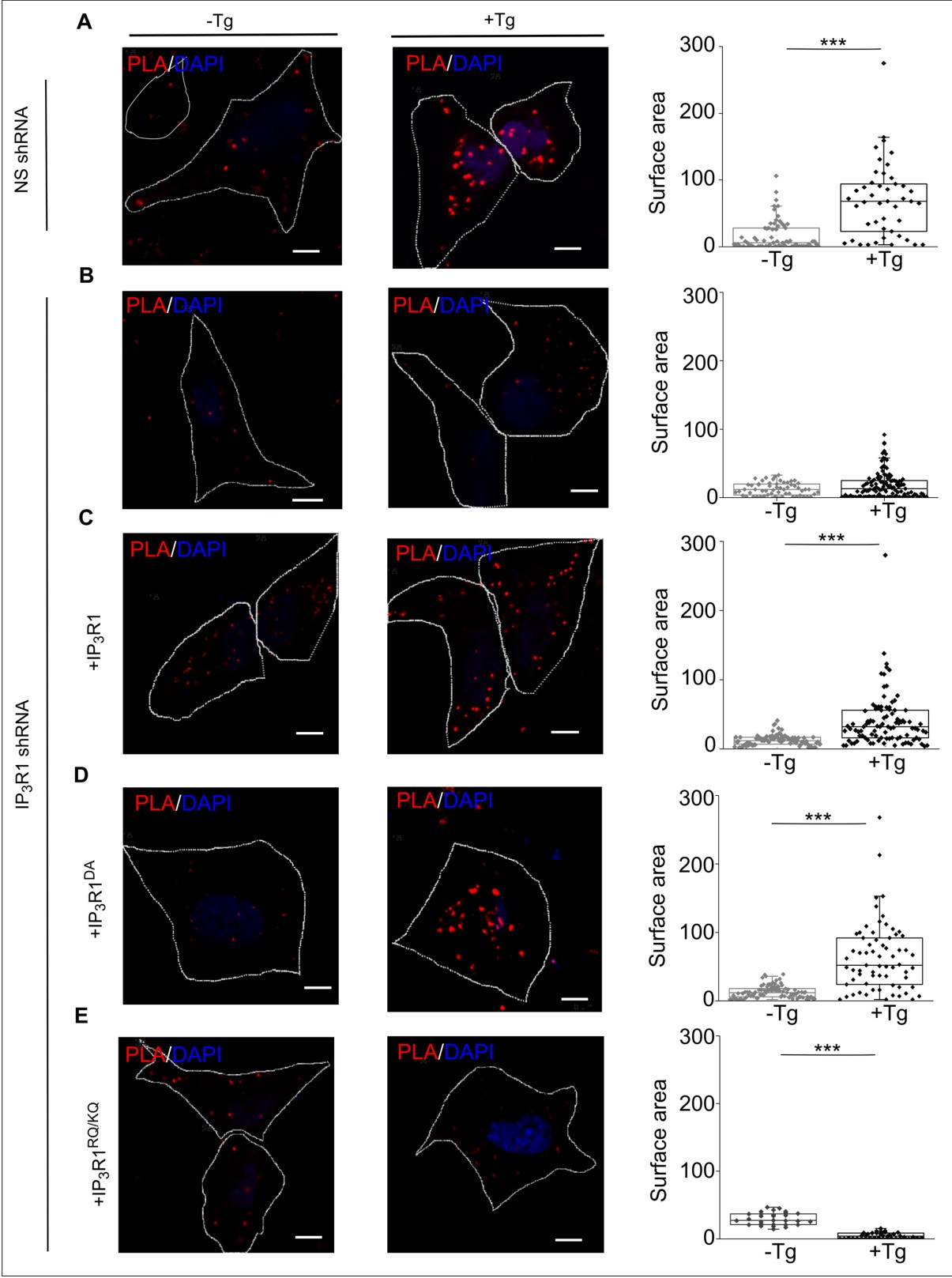

**Figure 5.** IP$_3$Rs promote interaction of STIM1 with Orai1. (**A–E**) PLA analyses of interactions between STIM1 and Orai1 in SH-SY5Y cells expressing NS-shRNA (**A**) or IP$_3$R1-shRNA alone (**B**) or with IP$_3$R1 (**C**), IP$_3$R1$^{DA}$ (**D**) or IP$_3$R1$^{RQ/KQ}$ (**E**). Confocal images are shown for control cells or after treatment with thapsigargin (Tg, 1 µM) in Ca$^{2+}$-free HBSS. PLA reaction product is red, and nuclei are stained with DAPI (blue). Scale bars, 5 µm. Summary results show

*Figure 5 continued on next page*

**Figure 5 continued**

the surface area of the PLA spots for 8–10 cells from two independent analyses. Individual values, median (bar) and 25th and 75th percentiles (box). ***p < 0.001, Student's *t*-test with unequal variances. See also *Figure 5—figure supplement 1*. Source data in *Figure 5—source data 1*.

The online version of this article includes the following source data and figure supplement(s) for figure 5:

**Source data 1.** IP$_3$Rs promote interaction of STIM1 with Orai1.

**Figure supplement 1.** Validation of PLA measurements of Orai1-STIM1 interactions.

**Figure supplement 1—source data 1.** Validation of PLA measurements of Orai1-STIM1 interactions.

it was modestly reduced in other studies of mammalian cells (*Bartok et al., 2019*; *Yue et al., 2020*). However, additional complexity is suggested by evidence that SOCE may be reduced in cells without IP$_3$Rs (*Chakraborty et al., 2016*; *Bartok et al., 2019*; *Yue et al., 2020*), by observations implicating phospholipase C in SOCE regulation (*Rosado et al., 2000*; *Broad et al., 2001*), by evidence that SOCE responds differently to IP$_3$Rs activated by different synthetic ligands (*Parekh et al., 2002*) and by some, albeit conflicting reports (*Woodard et al., 2010*; *Santoso et al., 2011*; *Béliveau et al., 2014*; *Sampieri et al., 2018*; *Ahmad et al., 2022*), that IP$_3$Rs may interact with STIM and/or Orai (*Woodard et al., 2010*; *Santoso et al., 2011*; *Béliveau et al., 2014*; *Sampieri et al., 2018*).

We identified two roles for IP$_3$Rs in controlling endogenous SOCE in human neurons. As widely reported, IP$_3$Rs activate STIM1 by releasing Ca$^{2+}$ from the ER, but they also, and independent of their ability to release Ca$^{2+}$, enhance interactions between active STIM1 and Orai1 (*Figure 8*). The second role for IP$_3$Rs can be supplanted by over-expressing other components of the SOCE complex, notably STIM1 or ESyt1 (*Figure 2K–M* and *Figure 7A and B*). It is intriguing that STIM1 (*Carrasco and Meyer, 2011*; *Lewis, 2020*), ESyt1 (*Giordano et al., 2013*) and perhaps IP$_3$Rs (through the IP$_3$-binding core) interact with phosphatidylinositol 4,5-bisphosphate (PIP$_2$), which is dynamically associated with SOCE-MCS (*Kang et al., 2019*). We suggest that the extent to which IP$_3$Rs tune SOCE in different cells is probably determined by the strength of Gq signaling, the proximity of IP$_3$Rs to nanodomains of PLC signaling and endogenous interactions between STIM1 and Orai1. The latter is likely to depend on the relative expression of STIM1 and Orai1 (*Woo et al., 2020*), the STIM isoforms expressed, expression of proteins that stabilize STIM1-Orai1 interactions (*Darbellay et al., 2011*; *Rana et al., 2015*; *Rosado et al., 2015*; *Knapp et al., 2022*), and the size and number of the MCS where STIM1 and Orai1 interact (*Kang et al., 2019*). The multifarious contributors to SOCE suggest that cells may differ in whether they express "spare capacity". In neuronal cells, loss of IP$_3$ (*Figure 4D*) or of the dominant IP$_3$R isoform (IP$_3$R1-shRNA; *Figures 1 and 2*) is sufficient to unveil the contribution of IP$_3$R to SOCE, whereas HEK cells require loss of both IP$_3$ and IP$_3$R1 to unveil the contribution (*Figure 4H and I*). The persistence of SOCE in cells devoid of IP$_3$Rs (*Figure 2—figure supplement 3D and E*; *Prakriya and Lewis, 2001*; *Ma et al., 2002*) possibly arises from adaptive changes within the SOCE pathway. This does not detract from our conclusion that under physiological conditions, where receptors through IP$_3$ initiate SOCE, IP$_3$Rs actively regulate SOCE.

The IP$_3$Rs that initiate Ca$^{2+}$ signals reside in ER immediately beneath the PM and alongside, but not within, the MCS where STIM1 accumulates after store depletion (*Thillaiappan et al., 2017*; *Figure 6A and B*). In migrating cells too, IP$_3$Rs and STIM1 remain separated as they redistribute to the leading edge (*Okeke et al., 2016*). Furthermore, there is evidence that neither STIM1 nor STIM2 co-immunoprecipitate with IP$_3$R1 (*Ahmad et al., 2022*). We suggest, and consistent with evidence that SOCE in cells without IP$_3$Rs can be restored by over-expressing E-Syt1 (*Figure 7A–C*), that ligand-bound IP$_3$Rs facilitate SOCE either by stabilizing the MCS wherein STIM1 and Orai1 interact, or by indirectly supporting STIM1 movement towards the MCS, rather than by directly regulating either protein. Stabilization of the MCS is analogous with similar structural roles for IP$_3$Rs in maintaining MCS between ER and mitochondria (*Bartok et al., 2019*) or lysosomes (*Atakpa et al., 2018*; *Figure 8*). Alternately, our observation that SOCE-dependent STIM1 movement to the MCS is reduced in presence of IP$_3$R1$^{RQ/KQ}$ (*Figure 6* and *Figure 6—figure supplement 1*), suggests that ligand-bound IP$_3$R1s could help in STIM1 mobilization to the MCS. The mechanism(s) by which ligand bound IP$_3$R1s might stabilize the MCS or stimulate STIM1 movement to the MCS remain to be elucidated by methods that can directly assay the MCS such as electron microscopy.

Since both contributions of IP$_3$Rs to SOCE require IP$_3$ binding (*Figure 3E and F*), each is ultimately controlled by receptors that stimulate IP$_3$ formation (*Figure 4B and C*). Convergent regulation by

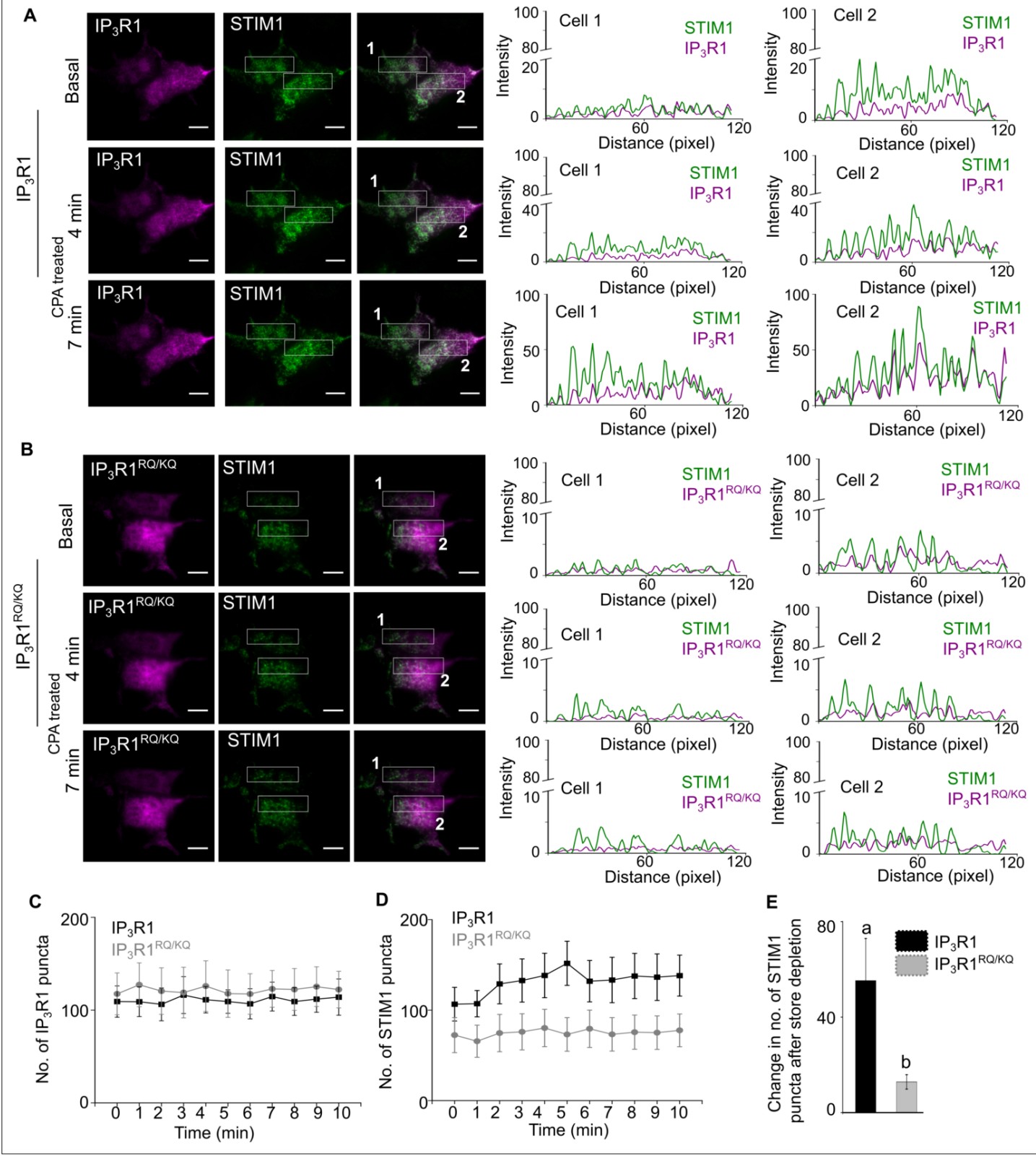

**Figure 6.** Ligand-bound IP₃R1 supports SOCE-dependent STIM1 movement to ER-PM contact sites. (**A–B**) Representative TIRF images of mVenus STIM1 co-transfected with either wild type mcherry-rat IP₃R1 (**A**) or IP₃R1$^{RQ/KQ}$ (ligand binding mutant), (**B**) in wild type SH-SY5Y cells before (Basal) and after CPA induced store depletion (CPA treated) at 4 min and 7 min. On the right are shown RGB profile plots of STIM1 (green) and IP₃R1, wild type or mutant (magenta) corresponding to the rectangular selections (Cell 1 and Cell 2). Scale bar is 10 μm.(**C–D**) Changes in number of IP₃R1 (**C**) and STIM1

*Figure 6 continued on next page*

*Figure 6 continued*

(**D**) puncta upon CPA-induced store depletion over a period of 10 min in the indicated genotypes. Mean ± s.e.m from seven cells from n=6 independent experiments. (**E**) Summary result (mean ± s.e.m) showing the change in the number of maximum STIM1 puncta formed after CPA-induced store depletion in the indicated genotypes. Mean ± s.e.m. of seven cells from n=6 independent experiments. Different letters indicate significant differences, p<0.05, Mann-Whitney U-test. See also *Figure 6—figure supplement 1*. Source data in *Figure 6—source data 1*.

The online version of this article includes the following source data and figure supplement(s) for figure 6:

**Source data 1.** Ligand-bound IP$_3$R1 supports SOCE-dependent STIM1 movement to ER-PM contact sites.

**Figure supplement 1.** Validation of fluorescent-tagged rat IP$_3$R1 constructs.

**Figure supplement 1—source data 1.** Validation of fluorescent-tagged rat IP$_3$R1 constructs.

IP$_3$Rs at two steps in the SOCE pathway may ensure that receptor-regulated PLC activity provides the most effective stimulus for SOCE; more effective, for example, than ryanodine receptors, which are also expressed in neurons (*Figure 8B*). By opening IP$_3$Rs parked alongside SOCE MCS (*Thillaiappan et al., 2017*; *Ahmad et al., 2022*), IP$_3$ selectively releases Ca$^{2+}$ from ER that is optimally placed to stimulate SOCE, and by facilitating Orai1-STIM1 interactions IP$_3$ reinforces this local activation of SOCE (*Figure 8A and B*).

We conclude that IP$_3$-regulated IP$_3$Rs regulate SOCE by mediating Ca$^{2+}$ release from the ER, thereby activating STIM1 and/or STIM2 (*Ahmad et al., 2022*) and, independent of their ability to release Ca$^{2+}$, IP$_3$Rs facilitate the interactions between STIM and Orai that activate SOCE. Dual regulation of SOCE by IP$_3$ and IP$_3$Rs allows robust control by cell-surface receptors and may reinforce local stimulation of Ca$^{2+}$ entry.

## Materials and methods
### Culture of human neural precursor cells

Human neural precursor cells (hNPCS) were derived from a human embryonic stem cell (hESC) line, H9/WA09 (RRID: CVCL_9773), using a protocol that inhibits dual SMAD signaling and stimulates Wnt signaling (*Reinhardt et al., 2013*) as described previously (*Gopurappilly et al., 2018*, 2019). hNPCs were grown as adherent dispersed cells on growth factor-reduced Matrigel (0.5%, Corning, Cat#356230) in hNPC maintenance medium (NMM) at 37 °C in humidified air with 5% CO$_2$. NMM comprised a 1:1 mixture of Dulbecco's Modified Eagle Medium with Nutrient Mixture F-12 (DMEM/F-12, Invitrogen, Cat#10565018) and Neurobasal medium (ThermoFisher, Cat#21103049), supplemented with GlutaMAX (0.5 x, Thermo Fisher, Cat#35050061), N2 (1:200, Thermo Fisher, 17502048), B27 without vitamin A (1:100, Thermo Fisher, Cat#12587010), Antibiotic-Antimycotic (Thermo Fisher, Cat#15240112), CHIR99021 (3 μM, STEMCELL Technologies, Cat#72052), purmorphamine (0.5 mM, STEMCELL Technologies, Cat#72202), and ascorbic acid (150 μM, Sigma, Cat#A92902). Doubling time was ~24 hr. Cells were passaged every 4–5 days by treatment with StemPro Accutase (Thermo Fisher, Cat#A1110501), stored in liquid nitrogen, and thawed as required. Cells were confirmed to be mycoplasma-free by monthly screening (MycoAlert, Lonza, Cat#LT07-318). hNPCs between passages 16 and 19 were used.

All experiments performed with hESC lines were approved by the Institutional Committee for Stem Cell Research, registered under the National Apex Committee for Stem Cell Research and Therapy, Indian Council of Medical Research, Ministry of Health, New Delhi.

### Stable knockdown of IP$_3$R1

An UltramiR lentiviral inducible shRNA-mir based on the shERWOOD algorithm (*Auyeung et al., 2013*; *Knott et al., 2014*) was used to inhibit IP$_3$R1 expression. The all-in-one pZIP vector, which allows puromycin-selection and doxycycline-induced expression of both shRNA-mir and Zs-Green for visualization, was from TransOMIC Technologies (Huntsville, AL). Lentiviral pZIP transfer vectors encoding non-silencing shRNA (NS, NT#3-TTGGATGGGAAGTTCACCCCG) or IP$_3$R1-targeting shRNA (ULTRA3316782- TTTCTTGATCACTTCCACCAG) were packaged as lentiviral particles using packaging (pCMV- dR8.2 dpvr, Addgene, plasmid #8455) and envelope vectors (pCMV-VSV-G, Addgene, plasmid #8454) by transfection of HEK293T cells (referred as HEK, ATCC, Cat# CRL-3216). Viral particles were collected and processed and hNPCs (passage 9) or SH-SY5Y cells were transduced

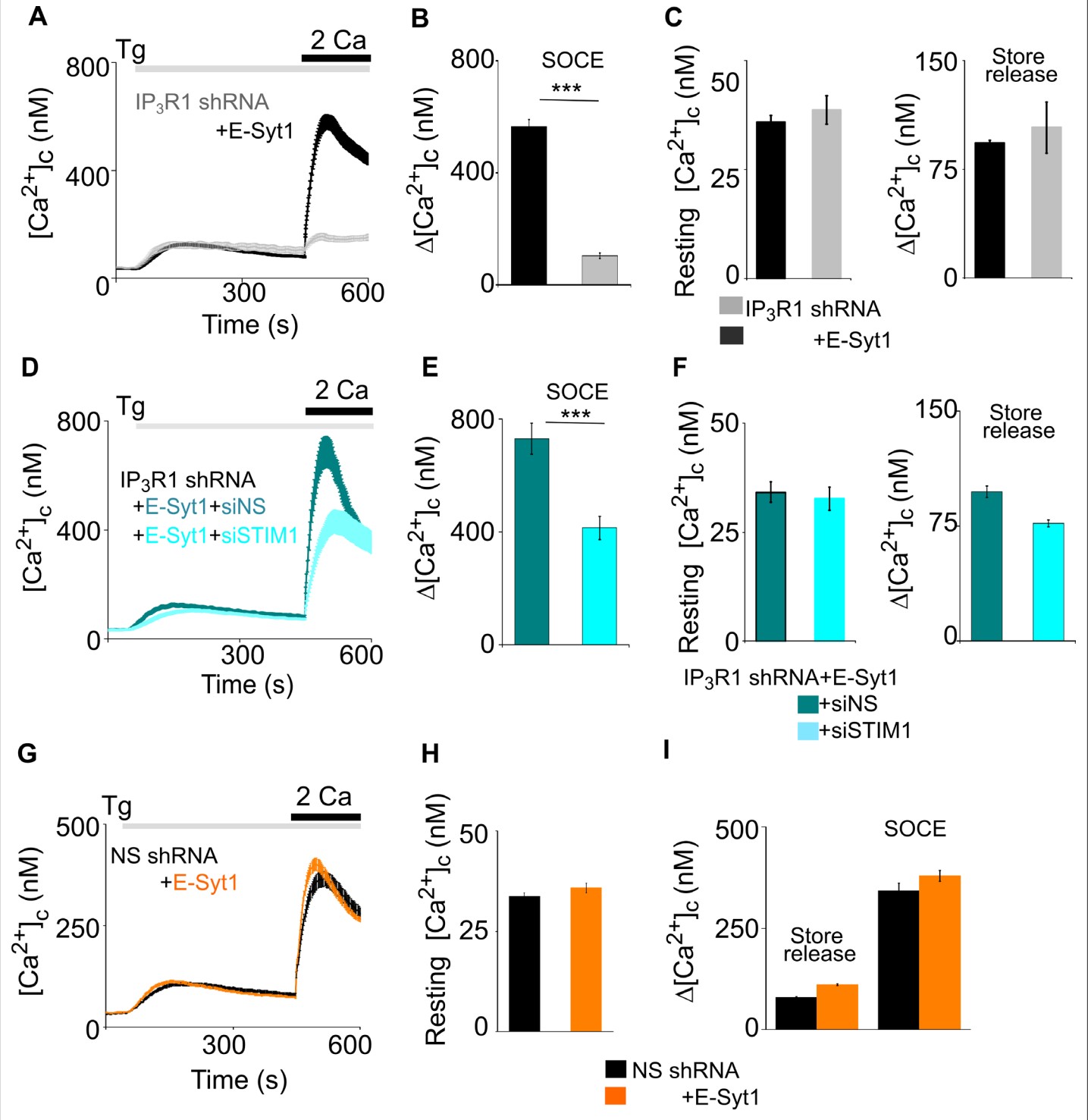

**Figure 7.** Extended synaptotagmins rescue SOCE in cells lacking IP$_3$R1. (**A**) SH-SY5Y cells expressing IP$_3$R1-shRNA alone or with E-Syt1 were stimulated with Tg (1 µM) in Ca$^{2+}$-free HBSS before restoring extracellular Ca$^{2+}$ (2 mM). Traces show mean ± s.e.m, for 20–80 cells from three experiments. (**B**) Summary results show Δ[Ca$^{2+}$]$_c$ evoked by restoring Ca$^{2+}$ (SOCE). Mean ± s.e.m, ***p < 0.001, Mann-Whitney U- test. (**C**) Summary results (mean ± s.e.m, n=20–80 cells) show resting [Ca$^{2+}$]$_c$ (left) and the peak Ca$^{2+}$ signals (Δ[Ca$^{2+}$]$_c$) evoked by thapsigargin (Tg, 1 µM) in Ca$^{2+}$-free HBSS for SH-SY5Y cells expressing IP$_3$R1-shRNA alone or with human E-Syt1. (**D**) Cells over-expressing E-Syt1 and treated with IP$_3$R1-shRNA in combination with either NS or STIM1 siRNA were stimulated with Tg (1 µM) in Ca$^{2+}$-free HBSS before restoration of extracellular Ca$^{2+}$ (2 mM). Mean ± s.e.m. from three experiments with 30–40 cells. (**E, F**) Summary results (mean ± s.e.m, n=30–40 cells) show SOCE evoked by Tg (**E**), resting [Ca$^{2+}$]$_c$ and the Tg-evoked Ca$^{2+}$ release from intracellular stores (**F**). ***p< 0.001, Mann-Whitney U- test. (**G**) Similar analyses of cells expressing NS shRNA alone or with human E-Syt1 and then treated

*Figure 7 continued on next page*

*Figure 7 continued*

with Tg (1 µM) in $Ca^{2+}$-free HBSS before restoring extracellular $Ca^{2+}$ (2 mM). Mean ± s.e.m. from three experiments with 115–135 cells. (**H, I**) Summary results (mean ± s.e.m, n=115–135 cells) show resting $[Ca^{2+}]_c$ (**H**) and $\Delta[Ca^{2+}]_c$ evoked by Tg (store release) or $Ca^{2+}$ restoration (SOCE) (**I**). No significant difference, Mann Whitney U-test. Source data in *Figure 7—source data 1*.

The online version of this article includes the following source data for figure 7:

**Source data 1.** Extended synaptotagmins rescue SOCE in cells lacking $IP_3R1$.

(multiplicity of infection, MOI = 10) using Lipofectamine LTX with PLUS reagent (Thermo Fisher, Cat#15338100). Cells were maintained in media containing doxycycline (2 µg/ml, Sigma, Cat# D3072) to induce shRNA expression, and puromycin to select transduced cells (1 µg/ml for hNPCs; 3 µg/ml for SH-SY5Y cells; Sigma, Cat# P9620). Cells were passaged 4–5 times after lentiviral transduction to select for stable expression of shRNAs.

## Derivation of neurons from hNPCs

Neurons were differentiated from hNPCs stably transduced with shRNA. hNPCs were seeded at 50–60% confluence in NMM on coverslips coated with poly-d-lysine (0.2 mg/ml, Sigma, Cat#P7280). After 1–2 days, the medium was replaced with neuronal differentiation medium, which comprised a 1:1 mixture of DMEM/F-12 with Neurobasal supplemented with B27 (1:100), N2 (1:200), GlutaMAX (0.5 x) and Antibiotic-Antimycotic solution. Medium was replaced on alternate days. Neurons were used after 15–20 days.

## Culture and transfection of SH-SY5Y cells

SH-SY5Y cells (ATCC, USA, Cat# CRL-2266) were grown on culture dishes in DMEM/F-12 with 10% fetal bovine serum (Sigma, Cat# F4135) at 37°C in humidified air with 5% $CO_2$. Cells were passaged every 3–4 days using TrypLE Express (ThermoFisher, Cat# 12605036) and confirmed to be free of mycoplasma. Cells expressing shRNA were transiently transfected using TransIT-LT1 reagent (Mirus, Cat# MIR-2300) in Opti-MEM (ThermoFisher, Cat# 31985062). Plasmids (250 ng) and/or siRNA (200 ng) in transfection reagent (1 µg/2.5 µl) were added to cells grown to 50% confluence on glass coverslips attached to an imaging dish. Cells were used after 48 hr. The siRNAs used were to human Orai1 (100 nM, Dharmacon, Cat# L-014998-00-0005) or non-silencing (NS, Dharmacon, Cat# D-495 001810-10-05), to human STIM1 (Santa Cruz Biotechnology, Cat# sc-76589) or NS (Santa Cruz Biotechnology, Cat# sc-37007). The expression plasmids were $IP_3R1$ (rat type 1 $IP_3R1$ in pcDNA3.2/V5DEST vector) (*Dellis et al., 2008*), rat $IP_3R1^{DA}$ (D2550 replaced by A in pcDNA3.2 vector) (*Dellis et al., 2008*), rat

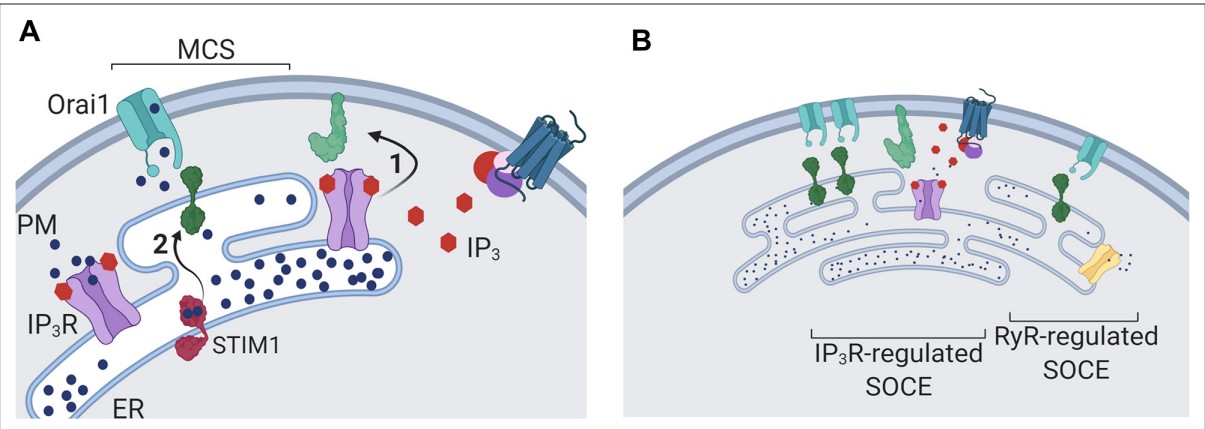

**Figure 8.** Dual regulation of SOCE by $IP_3Rs$. (**A**) SOCE is activated when loss of $Ca^{2+}$ from the ER, usually mediated by opening of $IP_3Rs$ when they bind $IP_3$, causes STIM to unfurl cytosolic domains (2). The exposed cytosolic domains of STIM1 reach across a narrow gap between the ER and PM at a MCS to interact with $PIP_2$ and Orai1 in the PM. Binding of STIM1 to Orai1 causes pore opening, and SOCE then occurs through the open Orai1 channel. We show that $IP_3Rs$ when they bind $IP_3$ also facilitate interactions between Orai1 and STIM, perhaps by stabilizing the MCS (1). Receptors that stimulate $IP_3$ formation thereby promote both activation of STIM (by emptying $Ca^{2+}$ stores) and independently promote interaction of active STIM1 with Orai1. (**B**) Other mechanisms, including ryanodine receptors (RyR), can also release $Ca^{2+}$ from the ER. We suggest that convergent regulation of SOCE by $IP_3R$ with bound $IP_3$ allows receptors that stimulate $IP_3$ formation to selectively control SOCE.

IP$_3$R1$^{RQ}$ (R568 replaced by Q of type 1 IP$_3$R in pCDNA3.2/V5DEST vector) (**Dellis et al., 2008**), rat IP$_3$R1$^{RQ/KQ}$ (R568 and K569 replaced by Q of type 1 IP$_3$R in pCDNA3.2/V5DEST vector), rat IP$_3$R1$^{1-604}$ (residues 1–604 of IP$_3$R with N-terminal GST tag in pCDNA3.2/V5DEST vector; **Dellis et al., 2008**), rat IP$_3$R3 (rat type 3 IP$_3$R in pcDNA3.2/V5DEST vector; **Saleem et al., 2013**), human mCherry-STIM1 (N terminal mCherry tagged human STIM1 in pENTR1a vector; **Nunes-Hasler et al., 2017**) and human extended synaptotagmin 1 (E-Syt1), a kind gift from Dr S. Muallem, NIDCR, USA (**Maléth et al., 2014**).

## CRISPR/Cas9 and Cas9n editing of SH-SY5Y cells

To allow either CRISPR/Cas9 or Cas9n-mediated disruption of IP$_3$R1 expression, we used a published method to clone gRNAs into the backbone vector (pSpCas9n(BB)–2A-Puro PX462 V2.0, Addgene, Cat#62987; **Ran et al., 2013**). Forward and reverse sgRNA oligonucleotides (100 µM) were annealed and ligated using T4 DNA ligase by incubation (10 µl, 37 °C, 30 min) before slow cooling to 20 °C. Plasmids encoding Cas9n were digested with *BbsI-HF* (37 °C, 12 hr), gel-purified (NucleoSpin Gel and PCR Clean-up kit from Takara) and the purified fragment was stored at –20 °C. A mixture (final volume 20 µl) of gRNA duplex (1 µl, 0.5 µM), digested px459 (for IKO null) or pX462 vector (for IKO) (30 ng), 10× T4 DNA ligase buffer (2 µl) and T4 DNA ligase (1 µl) was incubated (20 °C, 1 hr). After transformation of DH5-α competent *E. coli* with the ligation mixture, plasmids encoding Cas9 or Cas9n and the sgRNAs were extracted, and the coding sequences were confirmed (**Ran et al., 2013**). The plasmid (2 µg) was then transfected into SH-SY5Y cells (50–60% confluent) in a six-well plate using TransIT LT-1 reagent (Mirus Bio, Cat# MIR-2300). After 48 hr, puromycin (3 µg/ml, 72 hr) was added to kill non-transfected cells. IKO colonies were propagated and screened for Ca$^{2+}$ signals evoked by carbachol and for the presence of the IP$_3$R gene by genomic DNA PCR and droplet digital PCR using primers close to the region targeted by the gRNAs (**Miotke et al., 2014**). Three independently derived IKO lines, each with one residual IP$_3$R1 gene, were used for analyses of Ca$^{2+}$ signaling (see **Figure 2—figure supplement 1N-Q**). For one of the cell lines (IKO 2), disruption of one copy of the IP$_3$R1 gene was confirmed by genomic PCR, droplet digital PCR and western blotting (see **Figure 2—figure supplement 1K-M**). For the IKO null line, single-cell selection was done in a 96-well plate setup followed by screening for carbachol-evoked Ca$^{2+}$ signals from multiple clones. A single clone was selected (**Figure 2—figure supplement 1H**) and a western blot performed to confirm absence of IP$_3$R1 expression (**Figure 2—figure supplement 1F**). All the oligonucleotide sequences are described in **Supplementary file 1**.

## Plasmid construction

Mutagenesis and all DNA modifications were carried out using Q5 Hot Start high-fidelity 2 X Master Mix (New England BioLabs, Cat# M0494L) using the recommendations of the manufacturer. Primers used in this study (details given in **Supplementary file 1**) were synthesized by Integrated DNA Technologies (IDT). Mutations in the Ligand binding domain (R568Q and K569Q) of IP$_3$R1 were generated on the rat mCherry-IP$_3$R1 cDNA in pDNA3.1 Mutations in all the constructs were confirmed by sequencing.

## Ca$^{2+}$ imaging

Methods for single-cell Ca$^{2+}$ imaging were described previously (**Gopurappilly et al., 2019**). Briefly, cells grown as a monolayer (~70% confluence) on homemade coverslip-bottomed dishes were washed and loaded with Fura 2 by incubation with Fura 2 AM (4 µM, 45 min, 37 °C, Thermo Fisher, Cat# F1221), washed and imaged at room temperature in HEPES-buffered saline solution (HBSS). HBSS comprised: 20 mM HEPES, 137 mM NaCl, 5 mM KCl, 2 mM MgCl$_2$, 2 mM CaCl$_2$, 10 mM glucose, pH 7.3. CaCl$_2$ was omitted from Ca$^{2+}$-free HBSS. Treatments with carbachol (CCh, Sigma, Cat# C4382), thapsigargin (Tg, ThermoFisher, Cat# 7458), cyclopiazonic acid (CPA, Sigma Cat# C1530) or high-K$^+$ HBSS (HBSS supplemented with 75 mM KCl) are described in legends.

Responses were recorded at 2 s intervals using an Olympus IX81-ZDC2 Focus Drift-Compensating Inverted Microscope with 60×oil immersion objective (numerical aperture, NA = 1.35) with excitation at 340 nm and 380 nm. Emitted light (505 nm) was collected with an Andor iXON 897E EMCCD camera and AndoriQ 2.4.2 imaging software (RRID: SCR_014461). Maximal (R$_{max}$) and minimal (R$_{min}$) fluorescence ratios were determined by addition of ionomycin (10 µM, Sigma, Cat# 407953) in HBSS containing 10 mM CaCl$_2$ or by addition of Ca$^{2+}$-free HBSS containing BAPTA (10 mM, Sigma, Cat#

196418) and Triton X100 (0.1%). Background-corrected fluorescence recorded from regions of interest (ROI) drawn to include an entire cell was used to determine mean fluorescence ratios (R = $F_{340}/F_{380}$) (ImageJ), and calibrated to $[Ca^{2+}]_c$ from *Grynkiewicz et al., 1985*:

$$[Ca^{2+}]_c = K_D . F^{min}_{380} \Big/ F^{max}_{380} . (R - R_{min}) \Big/ (R_{max} - R)$$

where, $K_D$ = 225 nM (*Forostyak et al., 2013*).

## Western blots

Proteins were isolated in RIPA buffer (Sigma, Cat# R0278) with protease inhibitor cocktail (Sigma, Cat# P8340) or, for WB of Orai1, in medium containing 150 mM NaCl, 50 mM Tris, 1% Triton-X-100, 0.1% SDS and protease inhibitor cocktail. After 30 min on ice with intermittent shaking, samples were collected by centrifugation (11,000×*g*, 20 min) and their protein content was determined (Thermo Pierce BCA Protein Assay kit, ThermoFisher, Cat# 23225). Proteins (~30 µg/lane) were separated on 8% SDS-PAGE gels for IP$_3$R or 10% SDS-PAGE gels for STIM1 and Orai1, and transferred to a Protran 0.45 µm nitrocellulose membrane (Merck, Cat# GE10600003) using a TransBlot semi-dry transfer system (BioRad, Cat# 1703940). Membranes were blocked by incubation (1 hr, 20 °C) in TBST containing skimmed milk or bovine serum albumin (5%, Sigma, Cat# A9418). TBST (Tris-buffered saline with Tween) comprised: 137 mM NaCl, 20 mM Tris, 0.1% Tween-20, pH 7.5. Membranes were incubated with primary antibody in TBST (16 hr, 4 °C), washed with TBST (3 ×10 min), incubated (1 hr, 20 °C) in TBST containing HRP-conjugated secondary antibody (1:3000 anti-mouse, Cell Signaling Technology Cat# 7076 S; or 1:5000 anti-rabbit, ThermoScientific Cat# 32260). After 3 washes, HRP was detected using Pierce ECL Western Blotting Substrate (ThermoFisher, Cat# 32106) and quantified using ImageQuant LAS 4000 (GE Healthcare) and Image J. The primary antibodies used were to: IP$_3$R1 (1:1000, ThermoFisher, Cat# PA1-901, RRID: AB_2129984); β-actin (1:5000, BD Biosciences, Cat# 612656, RRID: AB_2289199); STIM1 (1:1000, Cell Signaling Technology, Cat# 5668 S, RRID: AB_10828699); Orai1 (1:500, ProSci, Cat# PM-5205, RRID: AB_10941192); IP$_3$R2 (1:1000, custom made by Pocono Rabbit Farm and Laboratory; *Mataragka and Taylor, 2018*); and IP$_3$R3 (1:500, BD Biosciences, Cat# 610313, RRID: AB_397705).

## Immunocytochemistry

After appropriate treatments, cells on a coverslip-bottomed plate were washed twice with cold PBS, fixed in PBS with paraformaldehyde (4%, 20 °C, 20 min), washed (3×5 min) with PBS containing Triton-X100 (0.1%, PBST) and blocked by incubation (1 hr, 20 °C) in PBST containing goat serum (5%). After incubation with primary antibody in PBST (16 hr, 4 °C) and washing with PBST (3×5 min), cells were incubated (1 hr, 20 °C) with secondary antibody in PBST containing goat serum, washed (3×5 min), stained (10 min, 20 °C) with DAPI (1 µg/ml in PBS; Sigma, Cat# D9542) and washed (5 min, PBST). Cells were then covered with glycerol (60% v/v) and imaged using an Olympus FV300 confocal laser scanning microscope with 20×or 60×oil-immersion objectives. Fluorescence was analyzed using ImageJ. The primary antibodies used were to: PAX6 (1:500, Abcam, Cat# ab195045, RRID: AB_2750924); Nestin (1:500, Abcam, Cat# 92391, RRID: AB_10561437); Ki67 (1:250, Abcam, Cat# ab16667, RRID: AB_302459); SOX1 (1:1000, Abcam, Cat# ab87775, RRID: AB_2616563); Tuj1 (βIII Tubulin 1:1000, Promega, Cat# G712, RRID: AB_430874); NeuN (1:300, Abcam, Cat# ab177487, RRID: AB_2532109); Doublecortin (1:500, Abcam, Cat# 18723, RRID: AB_732011); MAP2 (1:200, Abcam, Cat# ab32454, RRID: AB_776174); STIM1 (1:1000, Cell Signaling Technology, Cat# 5668 S, RRID: AB_10828699); and Orai1 (1:500, ProSci, Cat# PM-5205, RRID: AB_10941192).

## Proximity ligation Assay

The Duolink In Situ Red Starter Mouse/Rabbit kit was from Sigma (#Cat DUO92101) and used according to the manufacturer's protocol with primary antibodies to Orai1 (mouse 1:500) and STIM1 (rabbit 1:1000). Cells (~30% confluent) were treated with thapsigargin (1 µM, 5 min) in Ca$^{2+}$-free HBSS before fixation, permeabilization, and incubation with primary antibodies (16 hr, 4 °C) and the PLA reactants. Red fluorescent PLA signals were imaged using an Olympus FV300 confocal laser scanning microscope, with excitation at 561 nm, and a 60×oil-immersion objective. Quantitative analysis of the intensity and surface area of PLA spots used the 'Analyze particle' plugin of Fiji. Results are shown for

8–10 cells from two biological replicates of each genotype. Number of PLA spots in all genotypes and conditions were counted manually.

## Detection of STIM1 and IP$_3$R1 puncta using TIRF microscopy

SHSY5Y cells were cultured on 15 mm glass coverslips coated with poly-D-lysine (100 µg/ml) in a 35 mm dish for 24 h. Cells were co-transfected with 500 ng of mCherry rIP$_3$R1 and 200 ng mVenus STIM1 plasmids using TransIT-LT1 transfection reagent in Opti-MEM. Following 48 hr of transfection and prior to imaging, cells were washed with imaging buffer (10 mM HEPES, 1.26 mM Ca$^{2+}$, 137 mM NaCl, 4.7 mM KCl, 5.5 mM glucose, 1 mM Na$_2$HPO$_4$, 0.56 mM MgCl$_2$, at pH 7.4). The coverslips were mounted in a chamber and imaged using an Olympus IX81 inverted total internal reflection fluorescence microscope (TIRFM) equipped with oil-immersion PLAPO OTIRFM 60×objective lens/1.45 numerical aperture and Hamamatsu ORCA-Fusion CMOS camera. Olympus CellSens Dimensions 2.3 (Build 189987) software was used for imaging. The angle of the excitation beam was adjusted to achieve TIRF with a penetration depth of ~130 nm. Images were captured from a final field of 65 µm × 65 µm (300×300 pixels, one pixel = 216 nm, binning 2×2). Cells positive for both mCherry rIP$_3$R1 and mVenus STIM1 were identified using 561 nm and 488 nm lasers, respectively. The cells were incubated in zero calcium buffer (10 mM HEPES, 1 mM EGTA, 137 mM NaCl, 4.7 mM KCl, 5.5 mM glucose, 1 mM Na2HPO4, 0.56 mM MgCl2, at pH 7.4) for 2 min followed by addition of 30 µM CPA in zero calcium buffer. IP$_3$R1 and STIM1 puncta prior to CPA addition and after CPA addition were captured at 1 min intervals. Raw images were filtered for background correction and same setting was used across all samples. Regions where fresh STIM1 puncta (2–10 pixels) appeared post-CPA treatment at 10 mins were marked and subsequently IP$_3$R1 puncta (2–10 pixels) were captured from the same region. Change in the intensity of either STIM1 or IP$_3$R1 puncta was calculated from puncta of $\geq 2$ pixel by deducting the basal intensity at 0 min from the maximum intensity after CPA treatment using ImageJ ROI based mean grey value measurement. Particle analysis and RGB profile plot were done using ImageJ.

## Statistical analyses

All experiments were performed without blinding or prior power analyses. Independent biological replicates are reported as the number of experiments (n), with the number of cells contributing to each experiment indicated in legends. The limited availability of materials for PLA restricted the number of independent replicates (n) to 2 (each with 8–10 cells). Most plots show means ± s.e.m. (or s.d.). Box plots show 25th and 75th percentiles, median and mean (see legends). Where parametric analyses were justified by a Normality test, we used Student's $t$-test with unequal variances for two-way comparisons and ANOVA followed by pair-wise Tukey's test for multiple comparisons. Non-parametric analyses used the Mann-Whitney U-test. Statistical significance is shown by $^{***}p < 0.001$, $^{**}p < 0.01$, $^*p < 0.05$, or by letter codes wherein different letters indicate significantly different values (p<0.001, details in legends). All analyses used Origin 8.5 software.

Details of the plasmids and recombinant DNAs are given in ***Supplementary file 1***.

## Resource availability

### Lead contact

All requests for resources and reagents should be directed to the lead contact, Dr. Gaiti Hasan (gaiti@ncbs.res.in).

### Materials availability

Constructs and cell lines are available upon request. MTA required for cell lines.

## Acknowledgements

This research was supported by grants to GH from the Dept. of Biotechnology, Govt. of India (BT/PR6371/COE/34/19/2013) and NCBS-TIFR core support, to CWT from the Wellcome Trust (101844) and Biotechnology and Biological Sciences Research Council (BB/T012986/1) and to DIY from the NIH (NIDCR, DE014756). PC is supported by a DST-INSPIRE fellowship (DST/INSPIRE Fellowship/2017/IF170360) and she received an Infosys-NCBS travel award to visit CWT's lab at Cambridge. We are

grateful to Renjitha Gopurappilly (NCBS, TIFR) for the derivation of human neural precursor cells. We acknowledge use of the Central Imaging and Flow Cytometry Facility (CIFF), Stem Cell Culture Facility and Biosafety level-2 laboratory facility at NCBS, TIFR.

## Additional information

### Competing interests

Gaiti Hasan: Reviewing editor, eLife. The other authors declare that no competing interests exist.

### Funding

| Funder | Grant reference number | Author |
|---|---|---|
| Department of Science and Technology, Ministry of Science and Technology, India | DST/INSPIRE Fellowship/2017/IF170360 | Pragnya Chakraborty |
| Department of Biotechnology, Ministry of Science and Technology, India | BT/PR6371/ COE/34/19/2013 | Gaiti Hasan |
| Tata Institute of Fundamental Research | NCBS | Gaiti Hasan |
| Wellcome Trust | 101844 | Colin W Taylor |
| Biotechnology and Biological Sciences Research Council | BB/T012986/1 | Colin W Taylor |
| National Institutes of Health | DE014756 | David I Yule |
| Tata Institute of Fundamental Research | TIFR core support | Gaiti Hasan |

The funders had no role in study design, data collection and interpretation, or the decision to submit the work for publication. For the purpose of Open Access, the authors have applied a CC BY public copyright license to any Author Accepted Manuscript version arising from this submission.

### Author contributions

Pragnya Chakraborty, Conceptualization, Software, Formal analysis, Validation, Investigation, Visualization, Methodology, Writing – original draft, Writing – review and editing; Bipan Kumar Deb, Formal analysis, Validation, Methodology; Vikas Arige, Thasneem Musthafa, Sundeep Malik, Methodology; David I Yule, Conceptualization, Funding acquisition; Colin W Taylor, Gaiti Hasan, Conceptualization, Resources, Supervision, Funding acquisition, Writing – original draft, Writing – review and editing

### Author ORCIDs

Pragnya Chakraborty (ID) http://orcid.org/0000-0002-5916-5534
David I Yule (ID) http://orcid.org/0000-0002-6743-0668
Colin W Taylor (ID) http://orcid.org/0000-0001-7771-1044
Gaiti Hasan (ID) https://orcid.org/0000-0001-7194-383X

### Decision letter and Author response

Decision letter https://doi.org/10.7554/eLife.80447.sa1
Author response https://doi.org/10.7554/eLife.80447.sa2

## Additional files

### Supplementary files
• MDAR checklist

• Supplementary file 1. Details of plasmids and recombinant DNAs.

## Data availability

This study did not generate any computer code. The data supporting the findings of this study are available within the manuscript. All other data supporting the findings of this study are available in source data file of respective figures.

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
