## [Editor Report]

This paper proposes a fundamental new role for IP3 receptors in the regulation of store-operated calcium entry in neurons, in which IP3-bound receptors enhance the association of STIM1 and Orai1 independently of their ability to release Ca from the ER. While the evidence for this phenomenon is solid, experimental support for an underlying mechanism is incomplete and will require additional studies. The paper will appeal to cell biologists and neurobiologists interested in calcium signaling pathways, particularly store-operated calcium entry.

---

## [Decision Letter]

Decision letter after peer review:

Thank you for submitting your article "Regulation of Store-Operated ca^2+^ Entry by IP_3_ Receptors Independent of Their Ability to Release ca^2+^" for consideration by *eLife*. Your article has been reviewed by 3 peer reviewers, one of whom is a member of our Board of Reviewing Editors, and the evaluation has been overseen by Richard Aldrich as the Senior Editor. The following individuals involved in the review of your submission have agreed to reveal their identity: Khaled Machaca (Reviewer #2); Nicolas Demaurex (Reviewer #3).

Essential revisions:

– Are WT and IP3 binding deficient receptors recruited equivalently to membrane contact sites? This should be examined via high-resolution imaging.

– Are phosphoinositides involved in the recruitment of IP3R receptors?

– Does IP3R deficiency or mutant IP3R cause changes in the morphology/anatomy of the MCS? Is this independent of STIM1/Orai1? This should be addressed using high-resolution imaging and appropriate probes that do not perturb the contact sites.

– What is the dependence of IP3R occupancy with IP3 for SOCE activation? Does elevation of IP3 promote SOCE directly (independent of store-depletion)? This could be tested by examining what happens with CCh, the prediction is that this should further increase SOCE in response to TG (i.e. full depletion).

– Does STIM2 directly interact with IP3?

– What is the basis of the cell specificity of the IP3R effect in neuronal cells over HEK293 cells and immune cells where no effects on SOCE and CRAC currents were detected when all IP3Rs are deleted? This is a major point of difference from long-standing results and needs to be addressed.

Methodological issues:

– The conclusions regarding ER Ca levels and I3R activity are questionable. These should be improved using direct measures of ER [ca^2+^] using the now widely available ER targeted indicators.

– Determine if levels of STIM1 and Orai1 are altered in the IP3R1 KO cells.

– PLA studies should be done to show that YM and E-syt1 affect SOCE by modulating STIM-Orai interactions.

– PLA analysis is faulty and needs to be fixed.

– Orai1 Ab needs to be validated using an Orai1 KO HEK293 line.

In addition to the above points, the authors could consider addressing these additional questions as they would provide interesting additional insights in support of the paper's conclusions:

– Does IP3R deficiency alter (impair) phosphoinositide homeostasis at the PM?

– What is the phenotype of the full IP3R1 KO for MCS and Ca signaling?

– What about IP3R2 and 3 which are also present in neurons?

*Reviewer #1 (Recommendations for the authors):*

1. Based on the effects of YM, the authors suggest that only background levels of IP3 are required to maintain the normal amount of SOCE. Under these conditions, only a fraction of IP3R would be occupied. A prediction is that elevation of IP3 through a PLC-linked receptor should increase the occupancy of the IP3R and enhance SOCE in response to TG, even though TG by itself evokes full store depletion. This would strengthen support for the authors' model.

2. More direct evidence is needed to support the hypothesis that the IP3R increases STIM-Orai coupling and SOCE by stabilizing ER-PM junctions. MAPPER should be used to monitor the number and size of junctions after inhibitory treatments (IP3R1 KO, YM) or stimulatory treatments (CCh, IP3R1 restoration, STIM1 overexpression, E-Syt1). Because it links the ER and plasma membrane, a low expression of MAPPER may be required to avoid perturbing MCS formation. Perturbation would be indicated by rescue of SOCE by MAPPER in IP3R KO cells.

3. The cell specificity of the IP3R effects on SOCE is important. The STIM-Orai PLA experiments (or better yet, the MAPPER experiments in #2 above) should be done in HEK and SH-SY5Y cells in parallel. The prediction is that PLA and MCS will be reduced by IP3R KO in SH-SY5Y but not HEK cells. This would help establish a basis for explaining the cell-specific effects.

4. In Figure 4D, E, the effect of the Gq inhibitor on SOCE in NS shRNA cells is much less than in WT. What accounts for this difference?

5. As a control, the authors should confirm that STIM1 and Orai1 levels are not altered in the IP3R KO cells.

6. To show that YM and E-syt1 affect SOCE by modulating STIM-Orai interactions, PLA experiments should be done after YM inhibition of Gq, and after overexpression of E-syt1 as in Figure 5.

*Reviewer #2 (Recommendations for the authors):*

Were any controls performed to validate the specificity of the Orai1 antibody used to assess the expression levels of Orai1, as anti-Orai1 antibodies are notoriously non-specific? Has it been tested on Orai1-KO cells for example?

Line 198 should be Figure S6A.

*Reviewer #3 (Recommendations for the authors):*

1. IP3R is proposed to act as tethers at ER-PM junctions, yet no evidence is provided that WT and mutated receptors are recruited differentially to ER-PM junctions. To establish this point the authors should document the localization of the different receptors expressed (by immunofluorescence and PLA) to show that WT and mutant IP3R are differentially recruited to ER-PM contact sites and to show whether they localize near STIM and ORAI proteins.

2. The lack of an identified target for PM-bound IP3Rs is a significant limitation of the current study. While all the components of the STIM/ORAI machinery are potential targets the requirement for IP3 binding speaks for phosphoinositides. The authors should establish whether phosphoinositides are involved in the recruitment of IP3R receptors. Excellent tools are available to measure and manipulate the levels of phosphoinositides that can be used to document the effect of altered PIP2 levels on the localization of WT and mutated receptors.

3. The observation that the SOCE defect caused by IP3R deficiency can be rescued by the enforced expression of Esyt1 suggests that stabilizing membrane contact sites bypass the need for IP3R. This raises two questions: (1) Does IP3R deficiency indeed alters MCS structure or stability and (2) does IP3R deficiency alters MCS functions other than calcium signaling? ER-PM junctions are critical for the lipid replenishment of the PM thus the correction of the SOCE defect by ESyt1 could reflect changes in PM lipids rather than restored MCS tethering. The authors should provide morphological evidence that IP3R depletion and Esyt expression alter MCS and test whether the expression of lipid transport proteins (VAPs, NIRs, ORP) and of synthetic tethers such as MAPPERs can recapitulate the effect of ESyt1 on SOCE.

4. Why where the levels of the STIM2 protein not examined? STIM2 plays an important role in neurons and is thought to regulate and to be regulated by basal ER calcium levels. The lack of a ca^2+^ release channel is expected to increase the basal ca^2+^ levels within the ER, which in turn might prevent the activity of STIM2 and potentially reduce its expression levels. This control is particularly important since STIM2 was shown to interact with IP3R at membrane contact sites.

Methodological aspects:

5. As indicated in the public review I'm not convinced by the assays used to measure the filling state of intracellular calcium store. The authors report the amplitude of the calcium elevations evoked by thapsigargin in a ca^2+^-free medium to estimate the ca^2+^ content of intracellular stores. This parameter is considered to be equivalent in all the conditions presented despite some significant differences between conditions (e.g. Figure S5E, H, J). A better estimate would be provided by measuring the integrated response (area under the curve) preferably using non-calibrated traces as the non-linearity of the calibration augments variability. Inspection of the traces suggests that differences in ER ca^2+^ content are the norm rather than the exception because the averaged Tg responses differ visually between conditions in Figures 1H, 3B, C, E, S1D-E, S1K-L, S2K, S5K. The differentiated NPC shown in Figure 1H also appears to have a higher basal ca^2+^ which was not quantitatively evaluated. I would therefore ask the authors to re-analyse the data and to report the AUC of the Tg-evoked ca^2+^ responses. Given the importance of the ER ca^2+^ levels in controlling SOCE, the free ca^2+^ concentration within the ER should be determined for the critical conditions (control, IP3R KD, and reexpression of pore-dead and IP3-binding-deficient mutant). Excellent ratiometric cameleon indicators are available that allow quantitative recordings of [ca^2+^]ER. This is important because if the knockdown of IP3R impacts the resting ER ca^2+^ levels then the effects reported here might simply reflect changes in the activation levels of endogenous STIM proteins.

6. The PLA experiments are not analyzed properly. Reporting the total area of the dots is not common practice as these data are usually quantified by counting the number of dots per cell using Z stacks of 0.5µMm or less. The number of dots should be provided. The reference number for the STIM1 and ORAI1 antibodies used for PLA are not provided and their specificity should be validated by duolink using siRNA or KO for these proteins. These experiments should be repeated as an N of 2 biological replicates is too limited to support the conclusions.

7. Although CRISPR heterogeneous clones are validated by WB, I wonder why the authors did not present the sequences of change, or the impact. Also, I wonder why they decided to work with heterozygous and not with full KOs, which would give them a clean background. Is there any reason for choosing heterozygous over full KOs?

8. What about the other IP3R isoforms IP3R2 and 3? IP3R3 is predominant in neuronal systems but is not even discussed. Also, it is strange that its expression is not observed in the lines used in Figure 1. Is there any explanation for this? Do IP3R2 and 3 have a similar effect on MCS formation? Is the IP3 binding site conserved?

[Editors' note: further revisions were suggested prior to acceptance, as described below.]

Thank you for resubmitting your work entitled "Regulation of Store-Operated ca^2+^ Entry by IP_3_ Receptors Independent of Their Ability to Release ca^2+^" for further consideration by *eLife*. Your revised article has been evaluated by Richard Aldrich (Senior Editor), a Reviewing Editor, and three referees.

The manuscript has been improved, but all three reviewers noted that many of the essential revisions in the first review that called for experiments had not been done and that the rebuttal arguments given in their stead did not resolve the issues. Because of the potential importance of the main findings, we will consider a second revision, but only if the essential experiments to support the main findings can be performed. These include:

1) Are WT and IP3-binding-deficient mutant IP3Rs recruited equivalently to the MCS? This could be done by expressing labeled WT or binding-deficient mutant IP3R in IKO null cells or cells pretreated with IP3R shRNA.

2) Does IP3R deletion or expression of IP3-binding-deficient mutant IP3R cause changes in the number or dimensions of MCS? The previous review noted that a low level of MAPPER must be used in order to avoid perturbing the system and that rescue of SOCE by MAPPER would indicate too high a level (as was seen in the rebuttal data).

3) Does additional IP3 (from CCh or caged IP3) increase SOCE in cells with fully depleted stores (e.g., treated with TG)? The experiment using partial depletion with CPA (Figure 4) does not answer this question. The strong prediction of the model is that even with fully depleted stores (TG), increasing IP3 should increase SOCE.

4) How can the cell specificity of the IP3R effect on SOCE in neuronal cells over HEK293 and immune cells be explained? The results with SH-SY5Y cells are a major departure from longstanding results in other cell types that need to be addressed. The new data from HEK cells lack internal consistency and do not support the model proposed by the authors.

5) The PLA analysis should include both the number and area of spots.

Please see the individual reviews below for more detail on these essential points.

*Reviewer #1 (Recommendations for the authors):*

– Are WT and IP3 binding deficient receptors recruited equivalently?

They did not assess this. The rebuttal argument that "there is no reason to believe that IP3R mutants change their localization" does not apply, because they attribute an SOCE-enhancing function to IP3-bound receptors that cannot be filled with the non-binding mutant. So the question is whether the binding of IP3 generates this function by regulating localization.

– Does IP3R deficiency or mutant IP3R cause changes in the morphology/anatomy of the MCS?

They did not look at the effect of IP3R KO or KD on ER-PM contacts, stating that MAPPER would not work because it restores junctions on its own. This is certainly true for moderate/high expression levels (I am guessing those are the conditions for what they show in the rebuttal), but my original comment suggested specifically to look at low levels that may not perturb the number of junctions. In fact, MAPPER has been used to monitor changes in ER-PM junctions following store depletion with TG (Chang et al., Cell Rep 2013), and it seems well suited since they show an increased number of STIM-Orai PLA signals following the expression of IP3R. It may be a bit tricky to use properly but would not require as much effort as EM.

– What is the dependence of IP3R occupancy with IP3 for SOCE activation?

Their model predicts that increasing IP3 should increase SOCE even in cells treated with TG to fully deplete stores. This experiment was suggested, but they did not do it. The previous data (referenced in the rebuttal) used a partially depleting dose of CPA, and they conclude that IP3R enhances SOCE without enhancing store depletion. But this is an indirect argument that does not cleanly address the question. This is an important experiment to do because it is extremely unlikely that IP3R are saturated with IP3 at resting levels, and they see a large effect even with this low level of occupancy. Presumably, it would be even larger with higher IP3 levels. This would make the conclusion much more convincing in my view.

– What is the basis of the cell specificity of the IP3R effect in neuronal cells over HEK293 cells and immune cells where no effects on SOCE and CRAC currents were detected when all IP3Rs are deleted?

The new data from HEK cells about differences between neuronal/non-neuronal cells raise new questions. They state "In wild type or HEK-TKO cells, YM-254890 had no effect on thapsigargin-evoked SOCE, but it did inhibit SOCE in HEK cells lacking IP3R1" (lines 218-219). This does not make sense, as TKO cells lack IP3R1. Also, why would the addition of YM (and presumably reduction of basal IP3) reduce SOCE in cells lacking IP3R1, if it acts through IP3R1? These data do not seem self-consistent and do not make sense to me.

*Reviewer #2 (Recommendations for the authors):*

Cannot see the STIM2 WB in the supplemental figures? Figure 2 S1 is cropped and missing panels including the STIM2 WB.

The data for the inhibition of Cch-induced ca^2+^ release in the presence of YM is not shown in Figure 3 S1 as indicated in the text.

Figure 4D shows a dramatic inhibition of Tg-induced SOCE in the presence of YM in SH cells. This is quite surprising and not addressed, especially since with the ns shRNA the inhibition is much smaller (Figure 4E). It actually looks like SOCE inhibition in WT cells is more dramatic than the residual SOCE remaining after Ip3R1 shRNA treatment. This is quite confusing.

The new HEK293 data with YM to inhibit Gq is also confusing. Figure 4 S1 the panels are mislabeled and the statistical significance in the last panel is not clear. In TKO cells YM doesn't have any effect on SOCE, yet partial loss of IP3R1 (shRNA) with a reduction in IP3 levels decreases SOCE. It is concluded that loss of both IP3R and IP3 production is required to lower SOCE, yet both are lost in TKO cells with no reduction. Is the reduction in SOCE in TKO cells in the absence of YM compared to WT significant? This is not clear from the data in Figure S1F. Please clarify. As it stands conclusions in HEK293 cells are not justified.

Figure 2 S1 the panels are confusing: panel F is not described in the legend and for other panels, there is a mismatch between the Figure legend and figure. Also the full IP3R KO mentioned in the response to reviewers is not clear.

For the PLA analyses, the authors use the area of PLA spots as a measure of STIM-Orai interactions. They should also consider the number of spots as those as well reflect STIM1-Orai1 interactions. A more reliable way to encompass both would be to quantify the percent of cell footprint occupied by spots as a total measure of STIM1-Orai1 interactions.

Please carefully review the supplemental figures labelling and legends as it is currently difficult to follow.

*Reviewer #3 (Recommendations for the authors):*

This has been a difficult rebuttal to handle because the authors did not respond directly to my queries to explain why they did not consider the experiments suggested. To avoid ambiguities, I would appreciate it if they could clarify the points below and provide a point-by-point response to the initial queries

Location of receptors: I fail to understand the argument that "there is no reason to believe that IPTR mutants changed their localization". The receptors are proposed to facilitate SOCE by stabilizing contact sites (lines 285-286). Altered localization of mutated receptors can thus logically explain the signaling defect and in the absence of experimental evidence, we cannot exclude this possibility. Visualization of IPTR recruitment to STIM/ORAI clusters could be performed without tagging endogenous receptors, by PLA, or by TIRF imaging of re-expressed tagged receptors.

Role of phosphoinositides: The question here was whether altering PiP levels differentially impacts the recruitment of WT and mutated receptors to MCS, but this was not tested. Instead, the authors provide evidence that CCh-induced PIP2 depletion is enhanced in neuronal cells lacking IPTR1. This phenotype is consistent with a tethering function of IPTR1 that would facilitate PIP2 replenishment by stabilizing ER-PM contact sites but does not provide information as to whether phosphoinositides are involved in IPTR1 recruitment. Again, tagged receptors could be used to measure the impact of PIP2 depletion on their recruitment to the TIRF plane. Why was this not attempted?

Morphology of MCS: The new data show that MAPPER expression restores SOCE in IPTR1 null cells, confirming that stabilizing contact sites with artificial tethers corrects the signaling defect. Further evidence for a tethering function would require EM which is beyond the scope of this study.

Role of STIM2: A WB showing STIM2 levels is mentioned in the rebuttal as Figure 2 supplement 1M but this Figure is not included in the pdf provided. The key point here is whether the STIM2 levels are similar in WT and KO cells as the low intensity of bands in WB could reflect the poor affinity of the antibody.

Methodological issues:

ER ca^2+^ levels: The authors did not perform the suggested ER [ca^2+^] recordings, arguing that "they have not drawn conclusions regarding changes in ER-Ca" and that "there is no reason to believe that loss of IPTR1 affects ER-Ca". Yet in the Ms the Tg-evoked ca^2+^ release, an indicator of the ER ca^2+^ content, is repeatedly said to be unaltered or minimally perturbed (lines 99, 128, 132, 138, 141, 199, 229). The problem here is that, as detailed in the initial review, the fura-2 recordings show substantial differences in the amount of ca^2+^ mobilized from stores by thapsigargin. Please address the queries in point #5 of the first review regarding the re-analysis of the fura-2 data as differences in ER [ca^2+^] would impact the conclusions of the study.

PLA experiments: please provide the number of dots for each condition.

[Editors' note: further revisions were suggested prior to acceptance, as described below.]

Thank you for resubmitting your work entitled "Regulation of Store-Operated ca^2+^ Entry by IP_3_ Receptors Independent of Their Ability to Release ca^2+^" for further consideration by *eLife*. Your revised article has been evaluated by Kenton Swartz (Senior Editor) and a Reviewing Editor.

The manuscript has been improved but there are some remaining issues that need to be addressed, as outlined below:

While the revised manuscript is improved, several of the experiments that were requested by reviewers were not feasible or raised further questions, weakening the support for proposed mechanisms. However, the reviewers all agree that the main phenomenon described in the paper – a novel role for IP3R in modulating SOCE independent of their ability to release ca^2+^ – is exciting and worthy of publishing without further experiments to define an underlying mechanism. They also agree that given the uncertainty as to the mechanism, the authors will need to address the remaining points below clearly in the paper.

1) Are WT and IP3-binding-deficient mutant IP3Rs recruited equivalently to the MCS?

As Reviewer 2 noted, Figure 6A, B shows that the WT IP3R fluorescence in TIRF increases after CPA, but the RQ/KQ mutant does not. This does not appear to support the authors' conclusion that the two receptors are recruited equivalently to ER-PM junctions. The consensus view of the reviewers is that the time course of IP3R intensity in puncta after CPA should be plotted in Figure 6. This result has mechanistic implications, so it is important to show it clearly and discuss its interpretation in the paper.

2) Does IP3R deletion or expression of IP3-binding-deficient mutant IP3R cause changes in the number or dimensions of MCS?

The authors attempted to use MAPPER to monitor the number and size of MCS, but the lowest concentration of MAPPER DNA (200 ng) that produced detectable puncta also rescued SOCE in IKO null cells, suggesting that it altered the number of MCS by itself. It is surprising that 150 ng DNA did not label any MCS, while a slightly higher amount (200 ng) had a large enough effect on MCS stability to fully rescue the SOCE response. A more graded response would be expected, raising questions about whether MAPPER puncta at low transfection levels were overlooked. Nevertheless, difficulties in using MAPPER to track numbers and dimensions of native contact sites has been noted by other groups (including one of the reviewers). EM could be used to address this point, but is beyond the scope of the paper. Unfortunately, this means there is no direct evidence that IP3R increases the number of junctions, a key part of the hypothetical mechanism. For this reason, the reviewers agree that the authors should discuss the limitations of the available tools and propose alternative mechanisms (Discussion, around line 322). One such alternative would be that the IP3R directly interact with STIM1 rather than promoting junction formation. This might explain why the effects of YM in Figure 4 are so rapid (5 min), which may be too short a time for a profound loss of MCS.

3) Does additional IP3 (from CCh or caged IP3) increase SOCE in cells with fully depleted stores (e.g., treated with TG)?

(Figure 4 suppl 1A) Raising [IP3] with CCh after TG does increase the SOCE response but the effect is quite small, and as such does not offer strong support for the model. In fact, similar differences in peak Ca from SOCE were described as not significant in other experiments (e.g., Figure 2 supplement 3D). It would seem that if endogenous IP3 levels, which would be expected to only minimally occupy IP3R, are so potent at supporting SOCE (e.g., Figure 2E), raising IP3 significantly should cause a sizable increase in SOCE. It seems remarkable that resting [IP3], which is not enough to open the IP3R, can do so much, and even more so that the RQ mutant, with 10x lower affinity for IP3, rescues more than half the SOCE response (Figure 3E). It is difficult to imagine how this mutant would be binding any significant amount of IP3 in resting cells, unless it is sampling IP3 in a nanodomain close to PLC. Or perhaps as Reviewer 2 suggested, IP3Rs might be less important once STIM-Orai complexes are already formed. In any event, the small size of the CCh effect on SOCE needs to be acknowledged and discussed, as it appears to run counter to expectations given the hypothesis that IP3-bound receptors are needed for the effect.

4) How can the cell specificity of the IP3R effect on SOCE in neuronal cells over HEK293 and immune cells be explained?

The authors have provided a plausible explanation; however, there is no evidence that normal SOCE seen in TKO HEK cells "probably" arises from adaptive changes within the SOCE pathway (l. 310-311). "Possibly" would be more justified here (also considering that the response in TKO HEK cells appears somewhat reduced in Figure 2 Suppl 3D,E).

5) The PLA analysis should include both the number and area of spots.

The new data in Figure 5 suppl 1 show the number of PLA spots, but the quantification in panel O is confusing. The number of spots in the bar graph seems much lower than the number of spots visible in the PLA images of Figure 5 or the STIM/Orai puncta in Figure 5 suppl 1E-N. The authors should explain this apparent discrepancy (or choose more representative images to display).

*Reviewer #2 (Recommendations for the authors):*

For the new experiments shown in Figure 6 to address the recruitment of IP3R to MCS (ie the TIRF plane) the authors conclude that there is no difference in the recruitment of the WT vs RQ/KQ mutants. However, the intensity data for the two cells shows suggests otherwise: whereas there is a clear increase in the TIRF ROI for WT it is not apparent in the RQ/KQ mutant. Quantification of IP3R intensity in the TIRF plane on a per cell or ROI basis would quantitatively answer this question. It is clear that the number of IP3R puncta is not different between WT and the mutant (Figure 6C), but the intensity change is not clear given the way the data is presented. Need to show a time course of normalize intensity changes for WT and the mutant IP3R following CPA. This is important as it would argue for modulation of ER-PM MCS and as such provide a potential mechanism.

Figure 4—figure supplement 1A. Please elaborate on the finding of the relatively small increase with CCh compared to the significant decrease in SOCE following knockdown of IP3R (Figure 1D). This is an important finding as throughout the manuscript it is argued that ligand binding is critical for IP3R to support SOCE. Why the differential then with knockdown versus engaging the receptors after establishment of the STIM1-Orai1 complex? Are IP3Rs less important once the STIM1-Orai1 complexes are fully formed? Does TG treatment induce IP3 production?

*Reviewer #3 (Recommendations for the authors):*

Figure 6 shows TIRF images of mCherry-tagged receptors co-expressed with YFP-STIM1. The fluorescence pattern of the tagged receptors did not change appreciably during store depletion while additional STIM1 clusters appeared in cells expressing WT receptors. These data rely on overexpression with a substantial fraction of STIM1 pre-recruited in the TIRF plane that could explain the presence of the tagged IP3R in the TIRF plane, but nonetheless suggest that the WT and mutated receptors are not differentially recruited to contact sites. Whether changes in PPI impacts the distribution of receptors in the TIRF plane was not tested (using carbachol instead of CPA) but I agree that this experiment should be done with endogenously tagged receptors which would require considerable efforts. I thank the authors for addressing the points raised. The STIM2 levels are now documented and PLA data quantified. I have no further suggestions for changes.

---

## [Author Response]

Essential revisions:– Are WT and IP3 binding deficient receptors recruited equivalently to membrane contact sites? This should be examined via high-resolution imaging.

In this manuscript we have not attempted direct visualization of the IP_3_R at membrane contact sites. This is because previous publications from one of our groups (CWT) and others (Smith, Wiltgen and Parker, *Cell Calcium* 2009; Thillaiappan *et al.*, *Nat commun*. 2017; Lock *et al.*, *Sci. Signal* 2018) demonstrate that endogenous immobile clusters of IP_3_Rs that generate ca^2+^ puffs reside in ER-PM junctions alongside STIM whereas mobile IP_3_R clusters are present on the ER membrane. There is no reason to believe that IP_3_R1 mutants change their localization. Moreover, direct visualization of the IP_3_R1 mutants at MCS requires fluorescently tagged mutant and wild type IP_3_R1 by CRISPR editing that are currently not available in SH-SY5Y cells. Instead, we demonstrate that WT and IP_3_ binding mutant IP_3_R1s recruit SOCE molecules STIM1 and Orai1 differentially to the MCS (Figure 5). Possible roles for the IP_3_R1 and ligand binding during SOCE have been discussed in detail with relevant references to past work (Lines 291-311).

– Are phosphoinositides involved in the recruitment of IP3R receptors?

In order to attempt to answer this point we visualized an important PM phosphoinositide, PIP_2_, by expressing a PIP_2_ biosensor PH-PLCD1-GFP (Várnai and Balla, 2006 *Biochim. Biophys. Acta – Mol. Cell Biol. Lipids*) in SH-SY5Y cells lacking IP_3_R1 (Author response image 1, IKO null) and HEK cells lacking all three IP_3_Rs (TKO) (Author response image 1). In wild type SH-SY5Y cells a submaximal stimulus of carbachol (100µM) hydrolyzed plasma membrane (PM) bound PIP_2_ to approximately 60% of basal PM PIP_2_ whereas in IKO null cells PIP_2_ levels went down to approximately 35% of basal PM PIP_2_ after carbachol stimulation. Importantly, we did not observe a change in PM-localized PIP_2_ levels post-carbachol stimulation between HEK control and HEK-TKO cells. These data indicate higher PIP_2_ hydrolysis and/or reduced re-synthesis dynamics of membrane bound PIP_2_ in SH-SY5Y neuronal cells lacking IP_3_R1 but not in non-neuronal cells. We humbly submit that the relevance of altered PIP_2_ dynamics observed in IP_3_R1 knockout neuronal cells (Figure 1, below) to SOCE needs detailed future investigation involving genes that regulate PIP_2_ synthesis and hydrolysis. Our preliminary observation does not provide any new information in the context of our novel and important observation that ligand bound IP_3_Rs are the first step for initiating SOCE through STIM-Orai coupling at the ER-PM junction. Hence, we have not included these data in the manuscript.

**Author response image 1. sa2fig1:** Differential PIP2 dynamics in neuronal and non-neuronal cells. (A-B) Confocal images of cells expressing a PIP_2_ biosensor PH-PLCD1-GFP. SH-SY5Y (WT and IKO null) (A) and HEK- (WT and TKO) cells (B) before (basal) and after carbachol treatment (CCh, 100μM). Scale bar is 10µm. Summary results (mean+ s.e.m, from 3 independent experiments with a total of 15-18 cells) show the relative intensity of plasma membrane (PM) bound PH-PLCD1-GFP normalized to total fluorescence in their respective WT cells. Different alphabet indicate *P<0.01*, Student’s t-test with unequal variances.

– Does IP3R deficiency or mutant IP3R cause changes in the morphology/anatomy of the MCS? Is this independent of STIM1/Orai1? This should be addressed using high-resolution imaging and appropriate probes that do not perturb the contact sites.

This is an important point that we are unfortunately unable to answer due to the lack of appropriate probes that do not also affect ER-PM contact sites. Established methods by expression of MAPPER or Esyt1 will not work because both molecules restore ER-PM contact sites and rescue SOCE in IP_3_R1 shRNA or KO (IKO null) cells (Figure 6 for Esyt1 in IP_3_R1 shRNA cells) and see Author response image 2 for MAPPER in IP_3_R KO (IKO null) cells. The relationship of IP_3_Rs to MCS morphology has been discussed in detail (Lines 274-318).

– What is the dependence of IP3R occupancy with IP3 for SOCE activation? Does elevation of IP3 promote SOCE directly (independent of store-depletion)? This could be tested by examining what happens with CCh, the prediction is that this should further increase SOCE in response to TG (i.e. full depletion).

We examined the need for IP_3_ by partially depleting the ER of ca^2+^ using cyclopiazonic acid (CPA), a reversible inhibitor of SERCA, to allow submaximal activation of SOCE (Figure 3 —figure supplement 1M and 1N). Under these conditions, addition of carbachol in ca^2+^-free HBSS to cells expressing IP_3_R1-shRNA caused a small increase in [ca^2+^]_c_ (Figures 4A-4C). In the same cells expressing IP_3_R1^DA^, the carbachol-evoked ca^2+^ release was indistinguishable from that observed in cells without IP_3_R^DA^ (Figures 4B and 4C), indicating that the small response was entirely mediated by residual native IP_3_R1 and/or IP_3_R3. Hence, the experiment allows carbachol to stimulate IP_3_ production in cells expressing IP_3_R1^DA^ without causing additional ca^2+^ release. The key result is that in cells expressing IP_3_R1^DA^, carbachol substantially increased SOCE (Figures 4A-4C). We conclude that IP_3_, through IP_3_Rs, regulates coupling of empty stores to SOCE.

– Does STIM2 directly interact with IP3?

Western Blot analysis showed relatively low expression of STIM2 compared to STIM1 in SH-SY5Y cells (Figure 2 —figure supplement 1M) unlike non-neuronal cells (Ahmad *et al.*, *PNAS* 2022). Therefore we did not investigate STIM2 function and interactions in SH-SY5Y cells any further.

– What is the basis of the cell specificity of the IP3R effect in neuronal cells over HEK293 cells and immune cells where no effects on SOCE and CRAC currents were detected when all IP3Rs are deleted? This is a major point of difference from long-standing results and needs to be addressed.

We provide new data showing that in HEK cells, neither loss of IP_3_R1 nor inhibition of Gq/11 uncouples empty ca^2+^ stores form SOCE, but together they do uncouple (Figure 4). In agreement with these data, we now show that STIM-Orai interactions (visualized by PLA) are similar in HEK WT and HEK-TKO cells (Figure 2 —figure supplement 1F). The new results contribute to the expanded Discussion (p13-14) in which we suggest that multifarious regulation of SOCE is likely to provide different cells with different levels of ‘surplus capacity’ and so different susceptibilities to disabling single elements. Hence, in neurons, loss of IP_3_R is sufficient to inhibit SOCE, while HEK cells require loss of both IP_3_R and IP_3_. Our new results suggest that regulation of SOCE by IP_3_R is likely to be a widespread feature of mammalian cells.

Methodological issues:– The conclusions regarding ER Ca levels and I3R activity are questionable. These should be improved using direct measures of ER [ca^2+^] using the now widely available ER targeted indicators.

ca^2+^ measurements in this manuscript are all related to changes in cytosolic ca^2+^ either in response to store depletion (Thapsigargin) or an IP_3_ generating ligand, Carbachol. We have not drawn any conclusions regarding changes in ER-ca^2+.^ To the best of our knowledge there is no reason to believe that loss of IP_3_R1 (the predominantly expressed IP_3_R subtype in many mammalian neurons) affects ER-ca^2+^. There are small changes in ER-ca^2+^ when ALL three subtypes of IP_3_Rs are knocked out in either HEK cells or mouse fibroblasts. Our experiments have not addressed triple IP_3_R knock outs in neuronal cells. The study focuses on knockdown and knockout of IP_3_R1 alone.

– Determine if levels of STIM1 and Orai1 are altered in the IP3R1 KO cells.

As suggested we have performed the Westerns Blots (Figure S2M). The data show that levels of STIM1, STIM2 and Orai1 are not altered in IP_3_R1 KO (IKO null) cells.

– PLA studies should be done to show that YM and E-syt1 affect SOCE by modulating STIM-Orai interactions.

We have performed PLA analyses for YM treated wild type SH-SY5Y cells (Figure 4 —figure supplement 1A). They show reduced interaction between STIM1 and Orai1 after Tg-induced store depletion similar to our observation with IP_3_R1 KD SH-SY5Y cells (Figure 5). The ability of E-Syt1 to restore SOCE by enhancing STIM-Orai interactions by creating more ER-PM junctions is published (Chang *et al.*, *Cell Rep*, 2013; Giordano *et al.*, *Cell* 2013; Kang *et al.*, *Sci Rep.* 2019).

– PLA analysis is faulty and needs to be fixed.

Upon store depletion by thapsigargin the size of PLA spots grew significantly bigger due to recruitment of multiple STIM1 molecules to the ER-PM junctions and their interaction with more Orai1 molecules similar to what has been observed in non-excitable cells (Shen et al., *PNAS* 2021). The thapsigargin-mediated response was robust and significant (Figure 5). Hence, we have analyzed the surface area of each PLA spot to represent greater STIM1-Orai1 interactions at the MCS (Figure 5). In our studies we did not observe a significant change between the number of smaller PLA spots in resting cells vs the larger spots in thapsigargin-treated cells.

– Orai1 Ab needs to be validated using an Orai1 KO HEK293 line.

We validated the Orai1 Ab using siOrai1 expressing SH-SY5Y cells (Figure 3 —figure supplement 1F).

In addition to the above points, the authors could consider addressing these additional questions as they would provide interesting additional insights in support of the paper's conclusions.

In addition to the above points, the authors could consider addressing these additional questions as they would provide interesting additional insights in support of the paper's conclusions:– Does IP3R deficiency alter (impair) phosphoinositide homeostasis at the PM?

Yes, we have tested this and the results are described above (Figure 1 in this letter). While the results are indeed of interest they do not add to the conclusions of this manuscript. Hence, we have not included them here.

– What is the phenotype of the full IP3R1 KO for MCS and Ca signaling?

Full IP_3_R1 KO lines were made in SH-SY5Y cells and their phenotype is now included in Figure 2 —figure supplement 1. Their SOCE phenotype is similar to the single-copy IP_3_R1 knockout. However, the IP_3_R1 KO null cells were extremely fragile and grow very slowly. This is not surprising since IP_3_R1 is the predominant (~99%) IP_3_R isoform in SH-SY5Y cells. Hence, we have not used the complete knock out cells extensively in this study.

– What about IP3R2 and 3 which are also present in neurons?

IP_3_R3 overexpression rescues the SOCE phenotype of IP_3_R1 knockdown cells (Figure 2H). As shown in Figure 2A IP_3_R2 is not expressed in SH-SY5Y cells.

[Editors' note: further revisions were suggested prior to acceptance, as described below.]

The manuscript has been improved, but all three reviewers noted that many of the essential revisions in the first review that called for experiments had not been done and that the rebuttal arguments given in their stead did not resolve the issues. Because of the potential importance of the main findings, we will consider a second revision, but only if the essential experiments to support the main findings can be performed. These include:1) Are WT and IP3-binding-deficient mutant IP3Rs recruited equivalently to the MCS? This could be done by expressing labeled WT or binding-deficient mutant IP3R in IKO null cells or cells pretreated with IP3R shRNA.

As suggested we tested the recruitment of over-expressed mcherry-wild type IP_3_R1 and mcherry-IP_3_ binding deficient mutant (IP_3_R1^RQ/KQ^) to ER-PM junctions by TIRF microscopy in SH-SY5Y cells. Consistent with our PLA findings from Figure 5, we show that SOCE-dependent STIM1 translocation to the TIRF layer is significantly reduced in SH-SY5Y cells transfected with IP_3_R1^RQ/KQ^ as compared with cells transfected with WT IP_3_R1. In the same experiment we tested localisation of WT-IP_3_R1 and IP_3_R1^RQ/KQ^ to the TIRF layer upon store-depletion and SOCE. We see no change in EITHER IP_3_R1^WT^ or IP_3_R1^RQ/KQ^ localisations monitored for 10 minutes after store depletion (Figure 6), possibly due to overexpression of the constructs. Hence the second experiment suggested was not attempted. The current data do NOT support differential recruitment of WT and LBD IP_3_R1 to the MCS. Rather, as shown in Figure 8, they suggest that IP_3_ bound IP_3_Rs help create the MCS through interaction with as yet unidentified molecule(s) (see discussion lines 323-335).

2) Does IP3R deletion or expression of IP3-binding-deficient mutant IP3R cause changes in the number or dimensions of MCS? The previous review noted that a low level of MAPPER must be used in order to avoid perturbing the system and that rescue of SOCE by MAPPER would indicate too high a level (as was seen in the rebuttal data).

We have tried this experiment by transfecting 50ng, 150ng and 200ng of MAPPER. In SH-SY5Ycells we do not see MCS formation either with 50ng MCS or with 150ng of MAPPER (please see panel A in Author response image 3 for 150 ng). At 200ng we do see MCS but we also see rescue of SOCE. The concentrations used by Chang et al., Cell Rep 2013 of 15-50 ng MAPPER in HeLa cells appear not to work in SH-SY5Y cells. This is possibly because the ER-PM architecture of neuronal cells is different from non-neuronal cells like HeLa (PMID: 14493991; PMID: 30739879; PMID: 28559323).

**Author response image 3. sa2fig3:** 

3) Does additional IP3 (from CCh or caged IP3) increase SOCE in cells with fully depleted stores (e.g., treated with TG)? The experiment using partial depletion with CPA (Figure 4) does not answer this question. The strong prediction of the model is that even with fully depleted stores (TG), increasing IP3 should increase SOCE.

We have done the SOCE experiment in control SH-SY5Y cells by fully depleting stores using 2 µM thapsigargin (Tg) followed by 1µM carbachol (Cch) and we could see a small but significant potentiation of SOCE by addition of carbachol (Figure 4 —figure supplement 1A).

4) How can the cell specificity of the IP3R effect on SOCE in neuronal cells over HEK293 and immune cells be explained? The results with SH-SY5Y cells are a major departure from longstanding results in other cell types that need to be addressed. The new data from HEK cells lack internal consistency and do not support the model proposed by the authors.

We suggest that the extent to which IP_3_Rs tune SOCE in different cell types is determined by their “spare capacity” for SOCE. This is likely dependent on strength of Gq signaling, expression level and sub-cellular localisation of IP_3_R isoforms and interactions between STIM and Orai, based on cell specific regulators of SOCE (see discussion lines 300-315). In neuronal cells, loss of either IP_3_ (Figure 4D) or of the dominant IP_3_R isoform (IP_3_R1-shRNA; Figures 1 and 2) is sufficient to unveil the contribution of IP_3_R to SOCE, whereas HEK cells requires reduction in BOTH IP_3_ (+YM condition) and IP_3_R1 (IP_3_R1 shRNA) to unveil the contribution of ligand bound IP_3_Rs to SOCE (Figures 4H and 4I; Figure 4 - figure supplement 1E-1G ). The persistence of SOCE in HEK cells devoid of all three IP_3_Rs (HEK TKO; Figure 2 —figure supplement 3D and 3E) (Prakriya and Lewis, 2001; Ma et al., 2002) probably arises from adaptive changes within the SOCE pathway in the prolonged absence of all three IP_3_R subtypes. This does not detract from our conclusion that under physiological conditions, where receptors through IP_3_ initiate SOCE, IP_3_Rs actively regulate SOCE.

5) The PLA analysis should include both the number and area of spots.

We have included the number of PLA spots in Figure 5—figure supplement 1O. As discussed earlier, there is no difference in the number of PLA spots under +/-Tg treated condition. However, IP_3_R1 shRNA and IP_3_R1 shRNA+IP_3_R1^RQ/KQ^ cells had significantly few PLA spots compared to control shRNA cells.

Reviewer #1 (Recommendations for the authors):– Are WT and IP3 binding deficient receptors recruited equivalently?They did not assess this. The rebuttal argument that "there is no reason to believe that IP3R mutants change their localization" does not apply, because they attribute an SOCE-enhancing function to IP3-bound receptors that cannot be filled with the non-binding mutant. So the question is whether the binding of IP3 generates this function by regulating localization.

Response given in the essential review above. Please see Figure 6 in the revised manuscript.

– Does IP3R deficiency or mutant IP3R cause changes in the morphology/anatomy of the MCS?They did not look at the effect of IP3R KO or KD on ER-PM contacts, stating that MAPPER would not work because it restores junctions on its own. This is certainly true for moderate/high expression levels (I am guessing those are the conditions for what they show in the rebuttal), but my original comment suggested specifically to look at low levels that may not perturb the number of junctions. In fact, MAPPER has been used to monitor changes in ER-PM junctions following store depletion with TG (Chang et al., Cell Rep 2013), and it seems well suited since they show an increased number of STIM-Orai PLA signals following the expression of IP3R. It may be a bit tricky to use properly but would not require as much effort as EM.

A detailed response is given above in the essential review.

– What is the dependence of IP3R occupancy with IP3 for SOCE activation?Their model predicts that increasing IP3 should increase SOCE even in cells treated with TG to fully deplete stores. This experiment was suggested, but they did not do it. The previous data (referenced in the rebuttal) used a partially depleting dose of CPA, and they conclude that IP3R enhances SOCE without enhancing store depletion. But this is an indirect argument that does not cleanly address the question. This is an important experiment to do because it is extremely unlikely that IP3R are saturated with IP3 at resting levels, and they see a large effect even with this low level of occupancy. Presumably, it would be even larger with higher IP3 levels. This would make the conclusion much more convincing in my view.

Response given in the essential review above. Please see Figure 4 —figure supplement 1A.

– What is the basis of the cell specificity of the IP3R effect in neuronal cells over HEK293 cells and immune cells where no effects on SOCE and CRAC currents were detected when all IP3Rs are deleted?The new data from HEK cells about differences between neuronal/non-neuronal cells raise new questions. They state "In wild type or HEK-TKO cells, YM-254890 had no effect on thapsigargin-evoked SOCE, but it did inhibit SOCE in HEK cells lacking IP3R1" (lines 218-219). This does not make sense, as TKO cells lack IP3R1. Also, why would the addition of YM (and presumably reduction of basal IP3) reduce SOCE in cells lacking IP3R1, if it acts through IP3R1? These data do not seem self-consistent and do not make sense to me.

Reduced SOCE upon addition of YM is ONLY seen in HEK cells with IP_3_R1-shRNA (Figure 4H) and NOT in either HEK-WT (Figure 4 – supplement 1E) OR HEK TKO cells (Figure 4 – supplement 1F). We apologise for this confusion. Note that efficiency of knockdown with IP_3_R1-shRNA is not 100% (Figure 2A). These data are consistent with the idea that addition of YM reduces IP_3_ formation and thus further reduces ligand bound IP_3_R1 in IP_3_R1-shRNA HEK cells.

Reviewer #2 (Recommendations for the authors):Cannot see the STIM2 WB in the supplemental figures? Figure 2 S1 is cropped and missing panels including the STIM2 WB.The data for the inhibition of Cch-induced ca^2+^ release in the presence of YM is not shown in Figure 3 S1 as indicated in the text.Figure 4D shows a dramatic inhibition of Tg-induced SOCE in the presence of YM in SH cells. This is quite surprising and not addressed, especially since with the ns shRNA the inhibition is much smaller (Figure 4E). It actually looks like SOCE inhibition in WT cells is more dramatic than the residual SOCE remaining after Ip3R1 shRNA treatment. This is quite confusing.The new HEK293 data with YM to inhibit Gq is also confusing. Figure 4 S1 the panels are mislabeled and the statistical significance in the last panel is not clear. In TKO cells YM doesn't have any effect on SOCE, yet partial loss of IP3R1 (shRNA) with a reduction in IP3 levels decreases SOCE. It is concluded that loss of both IP3R and IP3 production is required to lower SOCE, yet both are lost in TKO cells with no reduction. Is the reduction in SOCE in TKO cells in the absence of YM compared to WT significant? This is not clear from the data in Figure S1F. Please clarify. As it stands conclusions in HEK293 cells are not justified.Figure 2 S1 the panels are confusing: panel F is not described in the legend and for other panels, there is a mismatch between the Figure legend and figure. Also the full IP3R KO mentioned in the response to reviewers is not clear.For the PLA analyses, the authors use the area of PLA spots as a measure of STIM-Orai interactions. They should also consider the number of spots as those as well reflect STIM1-Orai1 interactions. A more reliable way to encompass both would be to quantify the percent of cell footprint occupied by spots as a total measure of STIM1-Orai1 interactions.Please carefully review the supplemental figures labelling and legends as it is currently difficult to follow.

We apologise for these errors. The corrections had been made and revised supplementary figures have been uploaded.

Reviewer #3 (Recommendations for the authors):This has been a difficult rebuttal to handle because the authors did not respond directly to my queries to explain why they did not consider the experiments suggested. To avoid ambiguities, I would appreciate it if they could clarify the points below and provide a point-by-point response to the initial queriesLocation of receptors: I fail to understand the argument that "there is no reason to believe that IPTR mutants changed their localization". The receptors are proposed to facilitate SOCE by stabilizing contact sites (lines 285-286). Altered localization of mutated receptors can thus logically explain the signaling defect and in the absence of experimental evidence, we cannot exclude this possibility. Visualization of IPTR recruitment to STIM/ORAI clusters could be performed without tagging endogenous receptors, by PLA, or by TIRF imaging of re-expressed tagged receptors.

The experiment with tagged receptors is now shown in Figure 6. An explanation for the results are given in the essential review above.

Role of phosphoinositides: The question here was whether altering PiP levels differentially impacts the recruitment of WT and mutated receptors to MCS, but this was not tested. Instead, the authors provide evidence that CCh-induced PIP2 depletion is enhanced in neuronal cells lacking IPTR1. This phenotype is consistent with a tethering function of IPTR1 that would facilitate PIP2 replenishment by stabilizing ER-PM contact sites but does not provide information as to whether phosphoinositides are involved in IPTR1 recruitment. Again, tagged receptors could be used to measure the impact of PIP2 depletion on their recruitment to the TIRF plane. Why was this not attempted?

As evident in Figure 6 of the revised manuscript tagged receptors are present in the TIRF plane of resting SH-SY5Y cells BUT SOCE does not induce further recruitment of overexpressed tagged receptors to the TIRF plane. Probably, this experiment would needs to be done with endogenously tagged WT and mutant receptors. We have not attempted this as yet.

Morphology of MCS: The new data show that MAPPER expression restores SOCE in IPTR1 null cells, confirming that stabilizing contact sites with artificial tethers corrects the signaling defect. Further evidence for a tethering function would require EM which is beyond the scope of this study.Role of STIM2: A WB showing STIM2 levels is mentioned in the rebuttal as Figure 2 supplement 1M but this Figure is not included in the pdf provided. The key point here is whether the STIM2 levels are similar in WT and KO cells as the low intensity of bands in WB could reflect the poor affinity of the antibody.

STIM2 levels are not changed between WT and IKO cells. Please see the updated Supplementary Figure 2 (Figure 2- supplementary 1G).

Methodological issues:ER ca^2+^ levels: The authors did not perform the suggested ER [ca^2+^] recordings, arguing that "they have not drawn conclusions regarding changes in ER-Ca" and that "there is no reason to believe that loss of IPTR1 affects ER-Ca". Yet in the Ms the Tg-evoked ca^2+^ release, an indicator of the ER ca^2+^ content, is repeatedly said to be unaltered or minimally perturbed (lines 99, 128, 132, 138, 141, 199, 229). The problem here is that, as detailed in the initial review, the fura-2 recordings show substantial differences in the amount of ca^2+^ mobilized from stores by thapsigargin. Please address the queries in point #5 of the first review regarding the re-analysis of the fura-2 data as differences in ER [ca^2+^] would impact the conclusions of the study.

Changes in ER-ca^2+^ release vary among individual experiments to an extent. To control for this variation we have ALWAYS performed the control cell type (e.g NS shRNA) with the experimental condition in parallel in every experiment.

PLA experiments: please provide the number of dots for each condition.

This analysis has been added in Figure 5—figure supplement 1O.

[Editors' note: further revisions were suggested prior to acceptance, as described below.]

The manuscript has been improved but there are some remaining issues that need to be addressed, as outlined below:While the revised manuscript is improved, several of the experiments that were requested by reviewers were not feasible or raised further questions, weakening the support for proposed mechanisms. However, the reviewers all agree that the main phenomenon described in the paper – a novel role for IP3R in modulating SOCE independent of their ability to release ca^2+^ – is exciting and worthy of publishing without further experiments to define an underlying mechanism. They also agree that given the uncertainty as to the mechanism, the authors will need to address the remaining points below clearly in the paper.1) Are WT and IP3-binding-deficient mutant IP3Rs recruited equivalently to the MCS?As Reviewer 2 noted, Figure 6A, B shows that the WT IP3R fluorescence in TIRF increases after CPA, but the RQ/KQ mutant does not. This does not appear to support the authors' conclusion that the two receptors are recruited equivalently to ER-PM junctions. The consensus view of the reviewers is that the time course of IP3R intensity in puncta after CPA should be plotted in Figure 6. This result has mechanistic implications, so it is important to show it clearly and discuss its interpretation in the paper.

As suggested we measured the intensity of tagged and overexpressed WT IP_3_R1, IP_3_R1^RQ/KQ^ mutants as well as tagged STIM1 before and after CPA treatment within the ROIs containing visible STIM1 puncta. These data are in Figure 6 – supplement 1B-E, described in results on lines 256-264 and discussed in lines 326-336. There is a small increase in intensity of WT IP_3_Rs after CPA treatment that is not matched by an increase in puncta number. We do not see this change in RQ/KQ. Because the intensity change for IP_3_Rs is small, we are concerned that it may be an artefact of over-expression. We have stated that this result needs to be verified using alternate methods (lines 262-264).

2) Does IP3R deletion or expression of IP3-binding-deficient mutant IP3R cause changes in the number or dimensions of MCS?The authors attempted to use MAPPER to monitor the number and size of MCS, but the lowest concentration of MAPPER DNA (200 ng) that produced detectable puncta also rescued SOCE in IKO null cells, suggesting that it altered the number of MCS by itself. It is surprising that 150 ng DNA did not label any MCS, while a slightly higher amount (200 ng) had a large enough effect on MCS stability to fully rescue the SOCE response. A more graded response would be expected, raising questions about whether MAPPER puncta at low transfection levels were overlooked. Nevertheless, difficulties in using MAPPER to track numbers and dimensions of native contact sites has been noted by other groups (including one of the reviewers). EM could be used to address this point, but is beyond the scope of the paper. Unfortunately, this means there is no direct evidence that IP3R increases the number of junctions, a key part of the hypothetical mechanism. For this reason, the reviewers agree that the authors should discuss the limitations of the available tools and propose alternative mechanisms (Discussion, around line 322). One such alternative would be that the IP3R directly interact with STIM1 rather than promoting junction formation. This might explain why the effects of YM in Figure 4 are so rapid (5 min), which may be too short a time for a profound loss of MCS.

We agree and have made the necessary changes in the discussion (lines 331-336).

3) Does additional IP3 (from CCh or caged IP3) increase SOCE in cells with fully depleted stores (e.g., treated with TG)?(Figure 4 suppl 1A) Raising [IP3] with CCh after TG does increase the SOCE response but the effect is quite small, and as such does not offer strong support for the model. In fact, similar differences in peak Ca from SOCE were described as not significant in other experiments (e.g., Figure 2 supplement 3D). It would seem that if endogenous IP3 levels, which would be expected to only minimally occupy IP3R, are so potent at supporting SOCE (e.g., Figure 2E), raising IP3 significantly should cause a sizable increase in SOCE. It seems remarkable that resting [IP3], which is not enough to open the IP3R, can do so much, and even more so that the RQ mutant, with 10x lower affinity for IP3, rescues more than half the SOCE response (Figure 3E). It is difficult to imagine how this mutant would be binding any significant amount of IP3 in resting cells, unless it is sampling IP3 in a nanodomain close to PLC. Or perhaps as Reviewer 2 suggested, IP3Rs might be less important once STIM-Orai complexes are already formed. In any event, the small size of the CCh effect on SOCE needs to be acknowledged and discussed, as it appears to run counter to expectations given the hypothesis that IP3-bound receptors are needed for the effect.

The key point here is that neuronal cells have a certain SOCE capacity and to reach this they require ligand-bound IP_3_Rs. Our data do not support an infinite increase in SOCE in presence of higher and higher levels of IP_3_. In Figure 4 suppl 1A, the small change in SOCE by Cch addition is observed after maximal store depletion and in the presence of WT IP_3_R1. Under these conditions SOCE is already close to its maximal capacity. The excess IP_3_ generated in this condition thus results in a minimal change in SOCE. We have modified the text in lines 204 and 206 to reflect this better. As shown in Author response image 4 partial store depletion by a lower concentration of Tg results in reduced SOCE which can be further increased by addition of Cch. We did not include these data in the manuscript because reviewer 2 wanted us to try the experiment with complete store depletion. In Figure 3E SOCE is induced in the absence of WT IP_3_R1 and is thus very low. The RQ mutant is able to bring SOCE back to ~50% of its maximal value. We agree that it might well do this by the mechanism suggested by the reviewer (see lines 305-306).

**Author response image 4. sa2fig4:** 

4) How can the cell specificity of the IP3R effect on SOCE in neuronal cells over HEK293 and immune cells be explained?The authors have provided a plausible explanation; however, there is no evidence that normal SOCE seen in TKO HEK cells "probably" arises from adaptive changes within the SOCE pathway (l. 310-311). "Possibly" would be more justified here (also considering that the response in TKO HEK cells appears somewhat reduced in Figure 2 Suppl 3D,E).

We agree and have changed “probably” to “possibly”.

5) The PLA analysis should include both the number and area of spots.The new data in Figure 5 suppl 1 show the number of PLA spots, but the quantification in panel O is confusing. The number of spots in the bar graph seems much lower than the number of spots visible in the PLA images of Figure 5 or the STIM/Orai puncta in Figure 5 suppl 1E-N. The authors should explain this apparent discrepancy (or choose more representative images to display).

We have redone the quantification of PLA spots by counting them manually in Figure 5 supplement 1. We have also replaced images of PLA in Figure 5 with more representative images. In the previous version the PLA spots were counted through an automated mechanism which appeared to take two closely lying spots as one.

Reviewer #2 (Recommendations for the authors):For the new experiments shown in Figure 6 to address the recruitment of IP3R to MCS (ie the TIRF plane) the authors conclude that there is no difference in the recruitment of the WT vs RQ/KQ mutants. However, the intensity data for the two cells shows suggests otherwise: whereas there is a clear increase in the TIRF ROI for WT it is not apparent in the RQ/KQ mutant. Quantification of IP3R intensity in the TIRF plane on a per cell or ROI basis would quantitatively answer this question. It is clear that the number of IP3R puncta is not different between WT and the mutant (Figure 6C), but the intensity change is not clear given the way the data is presented. Need to show a time course of normalize intensity changes for WT and the mutant IP3R following CPA. This is important as it would argue for modulation of ER-PM MCS and as such provide a potential mechanism.

Please see our response to point 1 above.

Figure 4—figure supplement 1A. Please elaborate on the finding of the relatively small increase with CCh compared to the significant decrease in SOCE following knockdown of IP3R (Figure 1D). This is an important finding as throughout the manuscript it is argued that ligand binding is critical for IP3R to support SOCE. Why the differential then with knockdown versus engaging the receptors after establishment of the STIM1-Orai1 complex? Are IP3Rs less important once the STIM1-Orai1 complexes are fully formed? Does TG treatment induce IP3 production?

Please see our response to point 3 above. There is no reason to believe that Tg induces IP3 production.